# Angiopoietin-like proteins stimulate HSPC development through interaction with notch receptor signaling

Michelle I Lin[1], Emily N Price[1], Sonja Boatman[1], Elliott J Hagedorn[1], Eirini Trompouki[1], Sruthi Satishchandran[1], Charles W Carspecken[1], Audrey Uong[1], Anthony DiBiase[1], Song Yang[1], Matthew C Canver[1], Ann Dahlberg[2], Zhigang Lu[3,4], Cheng Cheng Zhang[4,5], Stuart H Orkin[1,6], Irwin D Bernstein[2], Jon C Aster[7], Richard M White[8,9,10], Leonard I Zon[1]*

[1]Stem Cell Program and Division of Hematology/Oncology, Howard Hughes Medical Institute, Boston's Children's Hospital and Dana Farber Cancer Institute, Harvard Medical School, Boston, United States; [2]Pediatric Oncology, Clinical Division, Fred Hutchinson Cancer Research Center, Seattle, United States; [3]Department of Physiology, University of Texas Southwestern Medical Center, Dallas, United States; [4]Department of Developmental Biology, University of Texas Southwestern Medical Center, Dallas, United States; [5]Department of Physiology and Developmental Biology, University of Texas Southwestern Medical Center, Dallas, United States; [6]Department of Pediatric Oncology, Dana Farber Cancer Institute, Boston, United States; [7]Department of Pathology, Brigham and Women's Hospital, Boston, United States; [8]Department of Cancer Biology, Memorial Sloan Kettering Cancer Center, New York, United States; [9]Department of Genetics, Memorial Sloan Kettering Cancer Center, New York, United States; [10]Department of Medicine, Memorial Sloan Kettering Cancer Center, New York, United States

*For correspondence: zon@enders.tch.harvard.edu

**Abstract** Angiopoietin-like proteins (angptls) are capable of ex vivo expansion of mouse and human hematopoietic stem and progenitor cells (HSPCs). Despite this intriguing ability, their mechanism is unknown. In this study, we show that angptl2 overexpression is sufficient to expand definitive HSPCs in zebrafish embryos. Angptl1/2 are required for definitive hematopoiesis and vascular specification of the hemogenic endothelium. The loss-of-function phenotype is reminiscent of the notch mutant mindbomb (mib), and a strong genetic interaction occurs between angptls and notch. Overexpressing angptl2 rescues mib while overexpressing notch rescues angptl1/2 morphants. Gene expression studies in ANGPTL2-stimulated CD34+ cells showed a strong MYC activation signature and myc overexpression in angptl1/2 morphants or mib restored HSPCs formation. ANGPTL2 can increase NOTCH activation in cultured cells and ANGPTL receptor interacted with NOTCH to regulate NOTCH cleavage. Together our data provide insight to the angptl-mediated notch activation through receptor interaction and subsequent activation of myc targets.

## Introduction

Human hematopoietic stem and progenitor cells (HSPCs) are defined as cells with the ability to self-renew and differentiate into all blood lineages. They provide tremendous therapeutic potential for bone marrow transplantation in the treatment of hematologic malignancy, inherited blood disorders, and cancer chemotherapy. An important goal in studying stem cells is to identify factors that can

**eLife digest** Bone marrow contains types of stem cell that can produce new blood and immune cells. Transplanting bone marrow from a healthy person can be used to treat people with certain disorders of the blood and immune system, by providing a new supply of regenerating bone marrow stem cells. Bone marrow transplants are also critical for individuals who had their own bone marrow stem cells destroyed by cancer or by toxic anti-cancer therapies like chemotherapy.

Acquiring bone marrow for transplants can be a difficult process. Sometimes doctors can use the patient's own stem cells for a transplant. But in circumstances where the patient lacks healthy bone marrow, a bone marrow donor must be found. Donors and recipients must be carefully matched: certain proteins on the donor's bone marrow cells must be very similar to the proteins on the recipient's bone marrow cells, or the recipient's immune system will attack and kill the new cells.

If scientists could learn to grow the stem cells found in bone marrow in laboratories, they could circumvent some of the challenges associated with bone marrow donation. To do that, scientists must first understand the precise molecular mechanisms that allow the blood cell-producing stem cells to regenerate themselves and produce new blood cells.

Angiopoietin-like proteins are commonly used to help stem cells grow in the laboratory, and Lin et al. have now looked in detail at how these proteins work. This revealed that angiopoietin-like proteins also cause the stem cells that produce blood cells to grow in zebrafish embryos and are necessary for the embryos' vascular system to develop properly.

Zebrafish embryos lacking angiopoietin-like proteins develop a similar set of developmental problems as zebrafish embryos with a mutation in another protein called Notch. Through a series of experiments, Lin et al. show that angiopoietin-like proteins interact with Notch and help transform Notch into its active form, which is necessary for blood stem cell growth. Lin et al. also found that both angiopoietin-like proteins and Notch affect the same signaling molecules. This suggests that the two proteins may work together as part of the same molecular pathway. The work suggests an alternative method to activate Notch for blood stem cell stimulation during processes such as bone marrow or cord blood transplantation.

expand HSPCs in vitro, while maintaining their self-renewal capacity. Scientists have turned to clues during embryonic hematopoietic development to identify novel factors that regulate this expansion. Developmental hematopoiesis occurs in at least two distinct phases, in which an initial transient wave produces mainly primitive erythrocytes and myeloid cells, followed by a definitive wave which produces long-term HSPCs (*Orkin and Zon, 2008*). Ontogeny studies across species have identified the aorta-gonad-mesonephros (AGM) region as the site from which definitive HSPCs arise (*Dieterlen-Lievre, 1975*; *Medvinsky and Dzierzak, 1996*; *Tavian et al., 1996*; *Jaffredo et al., 1998*). Recent studies using time-lapse imaging in live zebrafish embryos and live mouse thick tissue sections revealed that HSPCs bud off from the endothelium lining the ventral wall of the developing dorsal aorta (DA) prior to entering circulation (*Bertrand et al., 2010*; *Boisset et al., 2010*; *Kissa and Herbomel, 2010*). Subsequent to their birth in the AGM, HSPCs migrate to the fetal liver where they undergo significant expansion in vivo (*Ema and Nakauchi, 2000*) before colonizing the bone marrow and provide life-long supply of all blood cells.

Angiopoietin-like proteins (ANGPTLs) were recently identified as growth factors capable of expanding mouse (*Zhang et al., 2006*; *Zheng et al., 2011*; *Farahbakhshian et al., 2014*) and human (*Zhang et al., 2008*; *Khoury et al., 2011*; *Ventura Ferreira et al., 2013*; *Fan et al., 2014*) HSPCs in culture. ANGPTLs are secreted proteins that closely resembled the Angiopoietins, which are important vascular regulators, but despite structural similarities, they do not bind to TIE-2 or TIE-1 (*Kim et al., 1999*, *2000*; *Oike et al., 2004*). They also exert much wider functions outside of the vasculature such as regulation of lipid, glucose and energy metabolism (*Hato et al., 2008*), inflammation (*Tabata et al., 2009*), and cancer (*Zhu et al., 2011*).

Notch receptors (NOTCH1-4) are single-pass type I transmembrane receptors implicated in various developmental and disease processes including the activation of the hematopoietic program. They are synthesized as ~300 kD full-length precursor proteins that undergo a series of proteolytic

cleavages in order to become fully activated. Prior to translocating to the cell surface, NOTCH is first cleaved by furin-like convertases in the trans-Golgi compartment, resulting in a heterodimer composed of the N-terminal extracellular domain and the C-terminal-transmembrane/intracellular domain, bound through noncovalent linkage. Canonical NOTCH activation requires its obligatory interaction with NOTCH ligands belonging to the Delta/Serrate/Jagged/LAG-2 (DSL) family (*Fortini, 2009*; *Kopan and Ilagan, 2009*). Several studies support a model in which ligand binding followed by endocytosis creates a mechanical force that alters the conformation of a juxtamembrane NOTCH negative regulatory region (*Kopan and Ilagan, 2009*). This permits cleavage of NOTCH by ADAM/ TACE (a disintegrin and metalloprotease/tumor necrosis factor α converting enzyme) at Site 2 (S2), generating the membrane-anchored NOTCH extracellular truncation (NEXT) fragment. NEXT is a substrate for the γ-secretase complex, which cleaves Site 3 (S3), releasing the NOTCH intracellular domain (NICD) to freely translocate into the nucleus to interact with DNA binding protein CSL (CBF1/ Su(H)/Lag-1)/RBPjκ and initiate transcription of target genes.

While the role of NOTCH in adult HSPC homeostasis still remains controversial, NOTCH is irrefutably important during embryonic hematopoiesis. Several Notch receptors and ligands are expressed in the mouse AGM and deletion of *Notch1*, *Jagged1*, or *CSL* resulted in impaired intra-embryonic hematopoiesis (*Kumano et al., 2003*; *Robert-Moreno et al., 2005*, *2008*). *Notch* target genes such as *Gata2* (*Minegishi et al., 2003*), *Runx1* (*North et al., 2002*) and those belonging to the *Hairy* and *Enhancer-of-split* related basic helix-loop-helix transcription factors, *Hes1*, *Hey1*, and *Hey2*, are also expressed in the AGM (*Robert-Moreno et al., 2005*, *2008*; *Guiu et al., 2013*). Previous studies from our lab demonstrated that HSPC fate is dictated by the *notch-runx1* pathway, in which overexpression of *runx1* mRNA in the *notch* mutant *mindbomb (mib)* can partially restore the loss of HSPCs normally observed in *mib* (*Burns et al., 2005*). Furthermore, recent studies demonstrated an even earlier role for *notch* in which somite-derived signals such as *wnt16* (*Clements et al., 2011*) or physical intracellular contacts between the *jam1a/2a* adhesion proteins (*Kobayashi et al., 2014*) can regulate *notch* signaling in HSC precursors.

Because of their potential in hematological applications and therapy, it is important to decipher the molecular pathways on which these ANGPTLs act. Here, we utilized zebrafish genetics to help provide insights into the mechanism by which ANGPTLs can expand adult HSPCs. We found that *angptls1* and *2* are indispensible for zebrafish definitive hematopoiesis and that they genetically interacted with *notch* signaling. To further uncover potential mechanisms for this interaction, we utilized cultured human cells and found that ANGPTL2 mediates NOTCH receptor cleavage/activation, occurring at the level of ANGPTL receptor binding to NOTCH. Our novel findings that *angptls* can induce *notch* activation provide an additional layer of regulation of canonical *notch* signaling.

## Results

### Overexpression of *angptl2* increases definitive hematopoiesis

*Angptl2* and *3* are highly expressed in the mouse fetal liver during hematopoietic expansion (*Zhang et al., 2006*) but it is not known whether they are important prior to this. To determine the role of *zangptls* during zebrafish hematopoiesis, we first generated a stable heatshock-inducible transgenic (Tg) zebrafish overexpressing full-length *zangptl2* cDNA, *Tg(hsp70: zangptl2)*. Heatshocked embryos had increased *zangptl2* mRNA after 2 hr (*Figure 1—figure supplement 1A*). Definitive hematopoiesis in zebrafish embryos is assessed at 36 hr post-fertilization (hpf), when emerging HSPCs develop in the AGM marked by *cmyb* and *runx1* transcripts (*Burns et al., 2005*; *North et al., 2007*). We observed significantly higher number of *cmyb*- and *runx1*-positive HSPCs in heatshocked Tg embryos compared to their non-heatshocked or non-Tg siblings by whole mount in situ hybridization (WISH) (*Figure 1A*). Increased *cmyb*- and *runx1*-positive HSPCs were also found in the caudal hematopoietic tissue (or CHT, akin to mammalian fetal liver) and the pronephros, which matures into the kidney marrow, the mammalian bone marrow equivalent (*Figure 1A*). *Rag1*-positive differentiated thymic T-cells that derive from definitive HSPCs were increased in heatshocked Tg siblings (*Figure 1A*). Together, these results indicate that overexpression of *zangptl2* is sufficient to increase zebrafish definitive hematopoiesis in vivo, recapitulating the initial finding that ANGPTL2 can expand HSPCs ex vivo (*Zhang et al., 2006*).

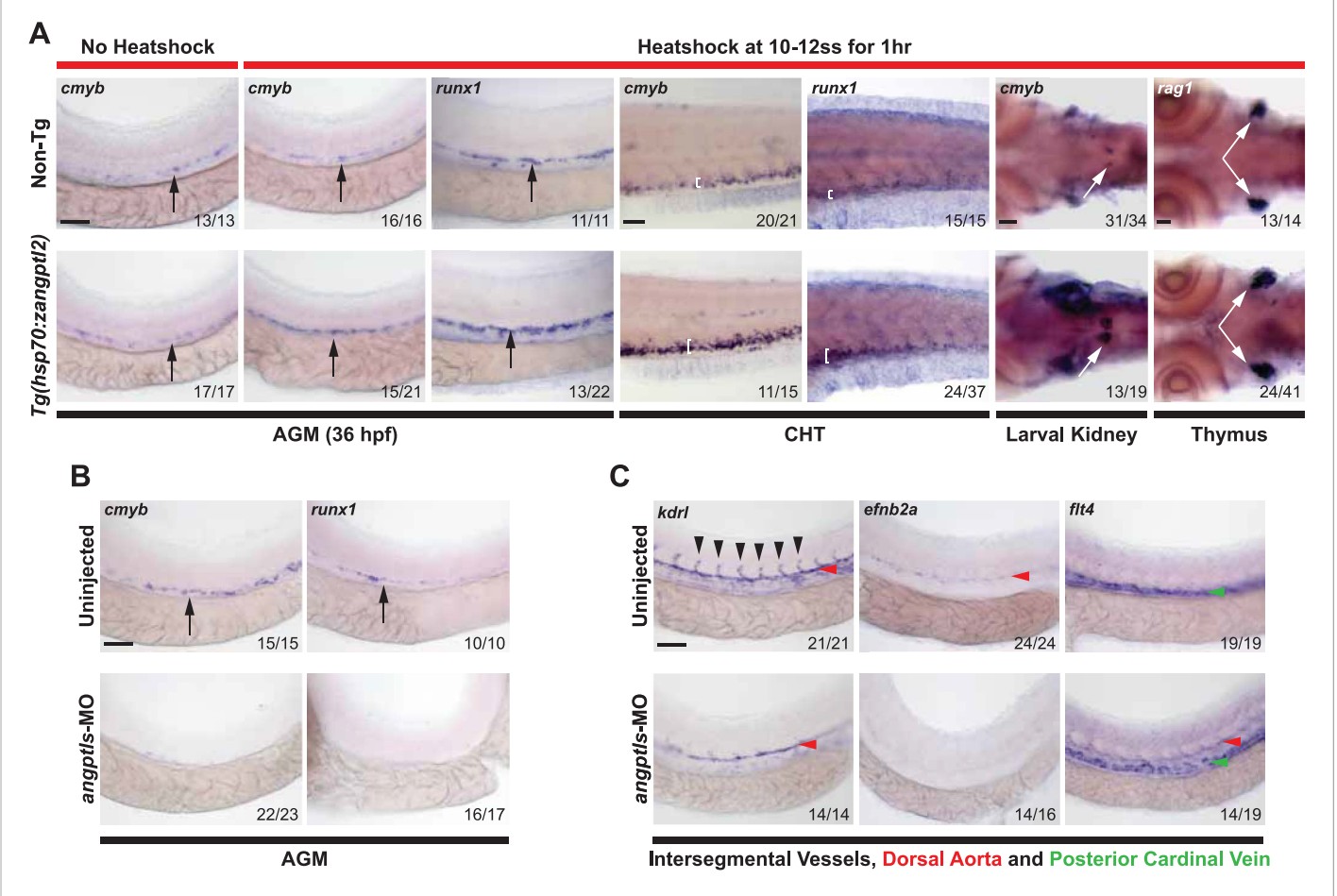

**Figure 1**. *Angptls* are sufficient and required for definitive hematopoiesis. (**A**) Heatshocked *Tg(hsp70:zangptl2)* embryos have increased *cmyb*- and *runx1*-positive HSPCs in the AGM (black arrows, 36hpf), CHT (white brackets, 3 days post-fertilization, dpf), larval kidney (white arrows, 4dpf) and *rag1*-positive T-cells in the thymus (white arrows, 5dpf) compared to control no heatshock or non-Tg siblings. (**B**) *Angptls*-MO morphants (2 ng) had decreased *cmyb*- and *runx1*-positive HSPCs (black arrows) in the AGM at 36hpf and (**C**) severe disruption to vascular development with loss of *kdrl*-positive ISVs (black arrowheads), loss of arterial *efnb2a* and ectopic expression of venous *flt4* in the DA (red arrowheads) in addition to PCV (green arrowheads) at 28hpf. Scale bars: 50 µm.

The following figure supplements are available for figure 1:

**Figure supplement 1**. *Zangptl2* overexpression in *Tg(hsp70:zangptl2)* embryos and endogenous *zangptl2* expression.

**Figure supplement 2**. *Angptls*-MO morphants had no defect in primitive hematopoiesis.

### *Angptl1* and *2* are required for definitive hematopoiesis and vascular specification

Previous studies demonstrated that *zangptl1* and *2* act cooperatively in zebrafish (*Kubota et al., 2005*). We next performed anti-sense knockdown experiments using previously established morpholinos (MOs) (*Kubota et al., 2005*) and found that while single *zangptl1*-MO or *angptl2*-MO can decrease *cmyb*-positive HSPCs in the AGM (data not shown), knocking down both *zangptls* (*angptls*-MO) led to a near complete absence of *cmyb*- and *runx1*-positive AGM HSPCs at 36hpf (*Figure 1B*), indicating that *zangptl1* and *2* are required for definitive HSPCs formation. In zebrafish, HSPCs arise from specialized *kdrl* (mammalian *KDR/Flk1* orthologue)-positive hemogenic endothelial cells in the dorsal aorta (DA) (*Bertrand et al., 2010*; *Kissa and Herbomel, 2010*). Because we observed the highest endogenous expression of *zangptl2* at ~23hpf (*Figure 1—figure supplement 1B*),

before the onset of definitive hematopoiesis, we examined the morphant vasculature at this time point. We found that angiogenic sprouting of *kdrl*-positive intersegmental vessels (ISVs) in control-MO injected (data not shown) or uninjected siblings was absent in *angptls*-MO morphants (*Figure 1C*). Axial vessel specification was also severely disrupted with decreased arterial *efnb2a* in the DA and ectopic expression of venous *flt4*, also in the DA, normally restricted to the posterior cardinal vein (PCV) by 28hpf (*Figure 1C*). These results suggest that *zangptls* regulation of definitive HSPC development may occur through an early specification of a patent and functional hemogenic endothelium.

To assess whether *zangptls* can act even earlier during primitive hematopoiesis, we examined *angptls*-MO morphants at 10–12 ss (somite stage, equivalent to ~12–14hpf) for defects in the bilateral stripes of the lateral plate mesoderm (LPM). Stage and somite-matched uninjected and *angptls*-MO injected siblings (marked by *myod*) had similar expression of early blood/vascular progenitor transcription factors (*Davidson and Zon, 2004*; *Dooley et al., 2005*; *Zhu et al., 2005*) such as *scl* (*Figure 1—figure supplement 2*), *lmo2*, and *fli1* (data not shown). Furthermore, *gata1*-positive primitive erythrocytes (*Detrich et al., 1995*) also appeared to be unchanged in the posterior LPM or intermediate cell mass at 24hpf (*Figure 1—figure supplement 2*), indicating that *zangptls1* and *2* are dispensable for primitive hematopoiesis.

## *Angptls* genetically interact with *notch*

The *angptls*-MO morphant phenotypes closely resembled that of the *notch* mutant, *mib* (*Lawson et al., 2001*; *Itoh et al., 2003*; *Burns et al., 2005*), which also exhibited defective definitive hematopoiesis and vascular specification (*Figure 2—figure supplement 1*). *Mib* encodes for the highly conserved E3 ubiquitin ligase important for endocytic processing of *notch* ligands (*Itoh et al., 2003*). To determine whether *angptls* and *notch* genetically interact, we injected *angptls*-MO into a *notch* reporter line, *Tg(Tp1bglob:eGFP)^um14* where eGFP is expressed under the control of a *notch*-responsive element consisting twelve *RBPjκ* binding sites (*Parsons et al., 2009*). Seen in *Figure 2A*, *angptls*-MO morphants had reduced eGFP expression, suggesting that *zangptls* are required for *notch* signaling. We then evaluated whether forced expression of constitutively active *notch* could rescue *angptls*-MO morphants. We observed that *angptls*-MO injected into heatshock-inducible double Tg, *Tg(hsp70:Gal4;UAS:NICD)* (*Scheer et al., 2001*; *Burns et al., 2005*) restored *cmyb*- and *runx1*-positive HSPCs (*Figure 2B* and data not shown). Surprisingly, *mib* embryos, normally devoid of HSPCs, crossed to *Tg(hsp70:zangptl2)* also rescued *cmyb*- and *runx1*-positive HSPCs (*Figure 2C* and data not shown) at 36hpf upon heatshock. Together, these genetic relationships between *zangptls* and *notch* place *zangptl* signaling downstream of *mib* but upstream of *notch* activation/*NICD* formation during zebrafish definitive hematopoiesis.

## *Notch* exert cell autonomous effects on hemogenic endothelial cells and HSPCs

While we know that ANGPTLs has direct effects on several mammalian cell types in culture (*Kubota et al., 2005*) but because they are secreted factors, the lack of their endogenous receptor identification in zebrafish limits our ability to assess which endogenous cell population they act upon. The mammalian ANGPTL receptor was identified to belong to a large superfamily of leukocyte immunoglobulin-like receptors (*Zheng et al., 2012*). Our attempts at establishing the zebrafish orthologue did not yield any likely candidate, due to the high similarity between the structural and functional domains for all members within this family and is thus beyond the scope of this paper. Despite this, we can extrapolate from whether *notch*, through which *zangptls* signal, can exert in a cell autonomous fashion on the endothelial cells or HSPCs during hematopoiesis. To do this, we utilized a transient overexpression system to image the AGM in live embryos by time-lapse spinning disk confocal microscopy. Because HSPCs in the AGM originate from the once *kdrl*-positive hemogenic endothelium, we first addressed whether *notch* has an effect on endothelial cells. We microinjected transposon-based vectors (Tol2) containing a *draculin* (*drl*) promoter driving eGFP and a 6.4-kb *kdrl* promoter driving expression of either constitutively active *NOTCH* (*NICD*, derived from human *NOTCH1*) or eGFP (as control) into *Tg(kdrl:Hras-mCherry);casper* embryos. The latter is a pan-vascular Tg line (*Chi et al., 2008*) that expresses membrane-bound mCherry in all endothelial cells and is bred into the

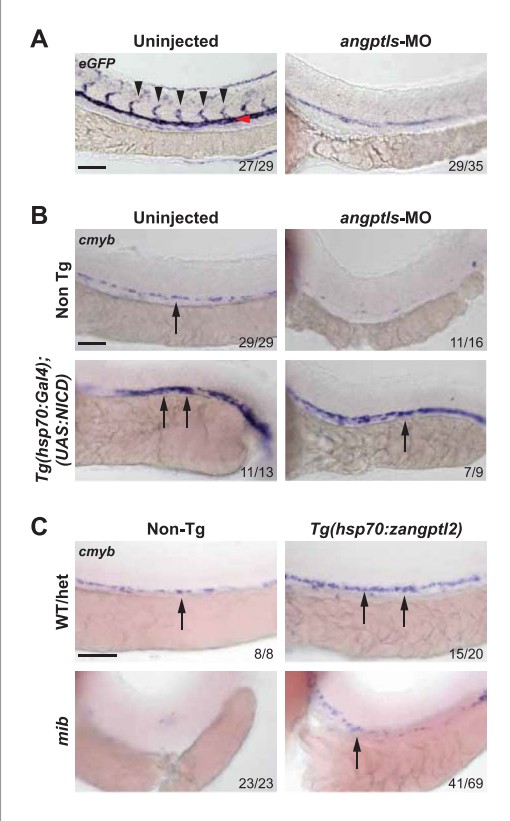

**Figure 2**. *Angptls* genetically interact with *notch*. (**A**) *Notch* eGFP Tg reporter embryos injected with *angptls*-MO (2 ng) had decreased *notch* activity in ISV (black arrowheads) and DA (red arrowhead) at 28hpf. (**B**) Overexpression of constitutively active *notch* in *angptls*-MO injected (2 ng) *Tg(hsp70:Gal4;UAS:NICD)* can restore *cmyb*-positive HSPCs (arrows) at 36hpf after heatshock. (**C**) *Mib* had no *cmyb*-positive HSPCs (arrows) at 36hpf compared to WT or het siblings. Overexpression of *zangptl2* in *mib* by crossing *Tg(hsp70:zangptl2)* into *mib* can restore this defect upon heatshock. Scale bars: 50 μm.
The following figure supplement is available for figure 2:

**Figure supplement 1**. *Notch* mutant *mib* has hematopoietic and vascular defects.

transparent *casper* (*White et al., 2008*) background for ease of imaging without interference from developing melanocytes. *Drl* is expressed very early (~3 ss) and marks all blood, vascular and cardiac lineages during zebrafish development (Mosimann C et al., submitted). Its sustained expression well before and throughout the onset of hematopoiesis would hence serve as a marker for all cells expressing the Tol2 vectors (GFP$^+$). We screened embryos with similar degree of GFP mosaicism to image from ~28hpf for 24 hr in the AGM and scored only the number of budding GFP$^+$ cells that lined the ventral floor of the DA (also mCherry$^+$). In the single frames (*Figure 3A*) of AGM budding time-lapse videos (*Videos 1 and 2*), we very rarely observe any AGM budding from GFP$^+$/mCherry$^+$ hemogenic endothelium in the control embryos (*Video 2*). In contrast, there was significantly more AGM budding events in embryos injected with constitutively active *NOTCH* (*Video 1*) (ranges for eGFP: 0–2; *NICD*: 0–6), all of which are depicted in the *Figure 3C* graph. These results highly suggest that NOTCH can exert a cell autonomous effect on hemogenic endothelial cells destined to become HSPC.

To further delineate whether *notch* also influences HSPCs, we did similar experiments as above by microinjecting Tol2 vectors containing *drl:eGFP* and a HSPC-specific *Runx1+ 23* enhancer to drive *NICD* or *eGFP* into *Tg (Runx1+23:NLS-mCherry);casper*. The Runx1 +23 enhancer element was identified in mouse to be highly expressed in the hematopoietic clusters in the AGM but unlike endogenous *Runx1*, it is expressed in only a very small subset of the underlying endothelial cells (*Nottingham et al., 2007*). The *Tg(Runx1 +23:NLS-mCherry)* Tg line expresses a nuclear localized (NLS) mCherry mainly restricted to HSPCs (*Tamplin et al., 2015*). We also detected nearly no mCherry mRNA expression in the hemogenic endothelial cells of

*Tg(Runx1+23:NLS-mCherry)* embryos at 36hpf by WISH (data not shown) justifying its use to drive expression of our transgenes specifically in budded HSPCs. Newly born HSPCs colonize the CHT starting at ~48hpf and expand tremendously until 80hpf in vivo. For these reasons, we scored the total number of transgene-positive (GFP$^+$) and mCherry$^+$ cells in the CHT at 72hpf to determine whether *NOTCH* could cell autonomously expand budded HSPCs. Shown in *Figure 3B*, we can readily observe significantly higher number of double positive HSPCs in which *NOTCH* is constitutively expressed compared to the control eGFP transgene injected siblings. These results are tabulated in *Figure 3D* and strongly indicated that *NOTCH* can exert cell autonomous effects on HSPCs. From these results in zebrafish, even though we cannot conclude whether *zangptls* themselves act cell autonomously on both endothelial cells and HSPCs, their downstream effector *notch* act cell autonomously.

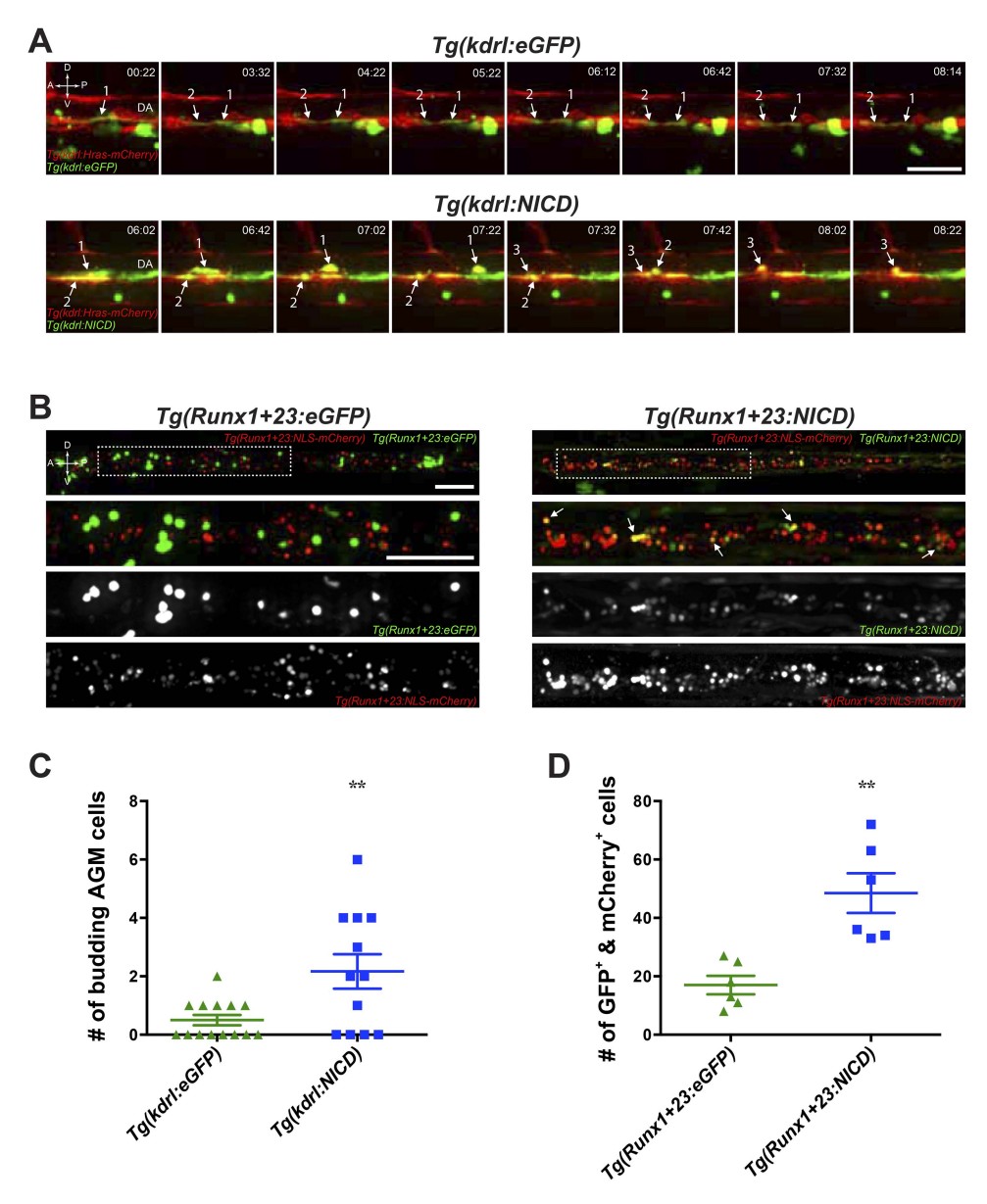

**Figure 3**. *Notch* cell autonomously increase definitive hematopoiesis. (**A**) Time-lapse sequence (hours:minutes post-28hpf) of HSPCs emerging from the ventral wall of the DA. Hemogenic endothelial cells (red) that have incorporated the injected transgene (green) are marked with numbers and white arrows. Each injected embryo was scored for 24 hr and tabulated in (**C**). Scale bar: 25 µm. (**B**) Still images of the CHT from 72hpf embryos. *Runx1*-positive HSPCs (red) that have incorporated the injected transgene (green) were scored for double positivity (yellow, examples marked by white arrows) and tabulated in (**D**). Boxed areas in the top panels are magnified and split into double or single fluorescent panels below. Scale bars: 50 µm. A: Anterior, P: Posterior, D: Dorsal; V: Ventral. Error bars denote S.E.M., **p < 0.01, compared to eGFP injected controls, Student's *t* test.

# *Myc* is an important downstream effector for *angptls* and *notch* signaling during hematopoiesis

To further assess the molecular signaling downstream of *zangptls* and *notch*, we examined a known target of *notch*, *myc*, and its relationship with respect to *zangptls*. The *MYC* proto-oncogene has been previously shown to be a direct *NOTCH* target in T-cell acute lymphoblastic leukemia (T-ALL) (*Palomero et al., 2006*; *Weng et al., 2006*). Moreover, *Myc* is also critical in the regulation of

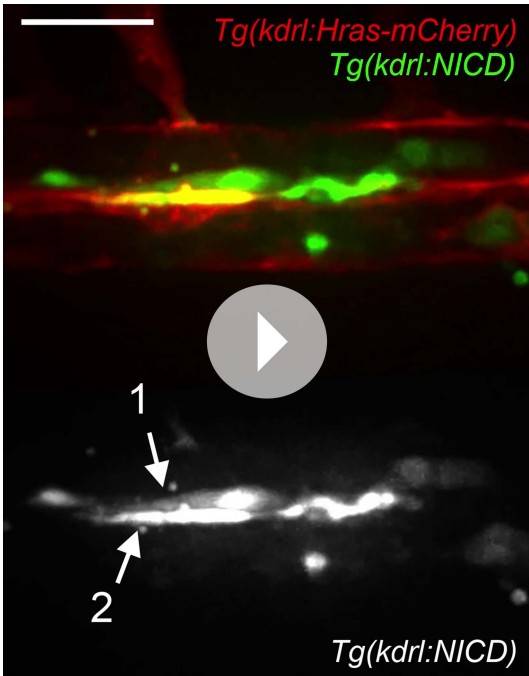

**Video 1.** Time-lapse video of AGM budding in embryos transiently expressing constitutively active *NOTCH*. Representative video depicting the HSPCs budding from the AGM in *Tg(drl:eGFP;kdrl:NICD)*-injected *Tg(kdrl:Hras-mCherry);casper*. Hemogenic endothelial cells (red) lining the ventral wall of the dorsal aorta that express the injected transgene (green) in the top panel depict at least three cell budding events (numbered with white arrow in the bottom panel showing GFP alone). Time-lapse imaging was captured at 10 min/frame and rendered at 3 frames/s. Each frame is a maximum projection of confocal z-stack, cropped from the entire AGM. Scale bar: 25 μm. DA: dorsal aorta; PCV: posterior cardinal vein.

hematopoietic/vascular development (*Wilson et al., 2004*; *Laurenti et al., 2008*). Conditional *cMyc* knockout in hematopoietic lineages resulted in severe cytopenia at E11.5 and lethality at E12.5 in mice (*He et al., 2008*), and *Myc*-deficient HSPCs are functionally defective, unable to engraft in recipient mice (*Wilson et al., 2004*). To first establish the role of *myc* during zebrafish definitive hematopoiesis, we performed morpholino knock-down of *myc* and observed a significant reduction in *cmyb*-positive HSPCs in the AGM (*Figure 4A*), suggesting its requirement. Next, to determine whether *myc* is downstream of *notch* during this process, we overexpressed *myc* in *mib* embryos and found partially restored *cmyb*-positive HSPCs in the AGM (*Figure 4B*). Our observed genetic interaction between *notch* and *angptls* in *Figure 2* prompted us to performed parallel *myc* rescue experiments in *angptl*-MO-injected morphants. We found similar restoration of *cmyb*-positive HSPCs in the AGM (*Figure 4C*), thereby implying that *myc* maybe one of the underlying effectors downstream of *zangptl* signaling. Together, these results further strengthened our hypothesis that *angptls* may exert their effects through *notch* signaling to converge upon *myc* during zebrafish definitive hematopoiesis.

## *Angptls* signaling activates *myc*

Our genetic data in zebrafish point to the possibility that *zangptls* signaling interacts with *notch*. We next carried out experiments in cultured cells to help dissect the mechanisms through which *zangptls* may act. First, to examine the molecular signaling downstream of ANGPTL, we performed gene expression studies on ANGPTL2-stimulated human CD34[+] cells using the Human Exon 1.0 ST Arrays. More than 3700 genes were differentially regulated with significant *q* value <0.05 (False Discovery Rate (FDR), Benjamini-Hochberg). In order to assess the biological significance of this signature, it was compared to 5562 gene sets in the Broad Molecular Signature Database using Gene Set Enrichment Analysis (GSEA). 284 gene sets showed a highly significant positive enrichment score (ES) with FDR<0.05, indicating that these sets contained genes that positively correlated with our expression phenotype, that is, upregulated in ANGPTL2-treated cells (*Supplementary file 1*). The top 20 gene sets were then chosen to perform a Leading Edge Analysis, in which the most significant subset of contributing genes from each gene set was cross-compared. We identified several clusters of genes/gene sets that contained direct *MYC* targets, contained a *MYC* signature, contained *MYC/MAX* binding motif, or were upregulated when *MYC* was upregulated. Despite a wide range of disease state responses or experimental parameters represented in these 284 gene sets, the strongest correlation was found with those relating to *MYC* signaling, an example of which is shown in *Figure 4D*. In concordance with the GSEA analysis, the ANGPTL2 signature was analyzed using Ingenuity Pathway Analysis (IPA), and *MYC* was again predicted to be one of the top upstream regulators in an active state (p-value of 2.89E-16). The compelling results from these bioinformatics analyses provided independent evidence to corroborate with our genetic data above, placing *myc* downstream of *zangptls* signaling during zebrafish hematopoiesis (*Figure 4C*).

To examine the effect of *MYC* downstream of NOTCH signaling, we performed chromatin immunoprecipitation using a NOTCH-specific antibody (*Wang et al., 2011*) followed by sequencing

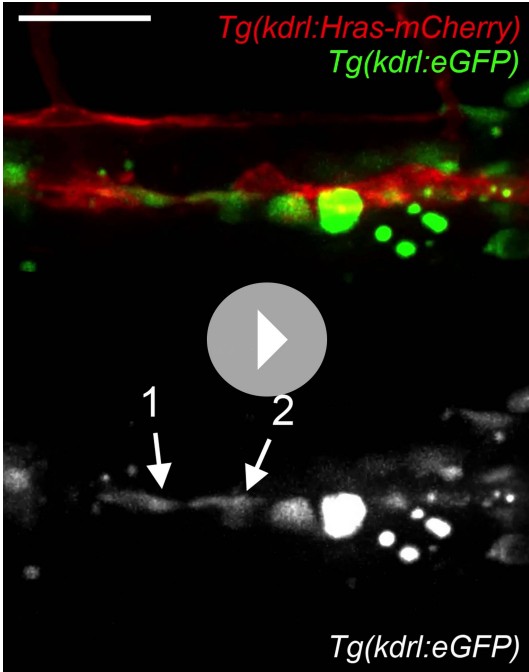

**Video 2.** Time-lapse video of AGM budding in control injected embryo. Representative video depicting the DA of control *Tg(drl:eGFP;kdrl:eGFP)* injected *Tg(kdrl:Hras-mCherry);casper*. In this example, the two cells that contain the control transgene never budded off from the AGM during the 24 hr imaged. Videos were captured on the same day as *NICD*-injected embryos.

(ChIP-seq) in human CD34[+] cells stimulated with ANGPTL2. Similar to the previously published ChIP-seq data (*Wang et al., 2011*), we found enrichment for *ETS, RUNX1,* and *ZNF143* motifs, the latter containing an embedded consensus sequence for *CSL/RBPjκ* (indicated by the red box in *Figure 4—figure supplement 1A*), in the sites bound by NOTCH. GREAT (Genomic Regions Enrichment of Annotations Tool) analysis revealed in some instances, that significantly more regions in our NOTCH ChIP-seq data set fall within the regulatory domains of *MYC* target genes or genes that are involved in the regulation of *MYC* targets (*Figure 4—figure supplement 1B*). Further path-way analyses using IPA revealed *MYC*-related genes forming the second highest ranking network as well as placing *MYC* as a significant upstream regulator in an active state (*Figure 4—figure supplement 1C*), again implying that those gene regions that were bound by NOTCH are indicative of *MYC* being positively regulated. GSEA analysis comparing the ANGPTL2 microarray signature to the NOTCH ChIP-bound *MYC* targets showed a strong and significant enrichment (*Figure 4E*), suggesting that the gene expression of *MYC* targets identified from the ANGPTL2 microarray overlapped significantly with those bound by NOTCH. Together with our genetic data and the fact that these informatics analyses revealed converging signaling on *MYC* activation by ANGPTL2 and NOTCH points to the likelihood that ANGPTL2 signaling may affect NOTCH activation.

## ANGPTL2 can induce NOTCH activation via receptor cleavage

To examine whether ANGPTLs have direct effects on NOTCH, we stimulated human CD34[+] progenitor cells with ANGPTL2 and observed a rapid increase in NOTCH receptor cleavage, generating the product NICD without affecting total levels of full-length NOTCH receptor (*Figure 5A*). Upon NOTCH cleavage, NICD can translocate into the nucleus to initiate transcription of target genes such as *HES1, RUNX1,* and *cMYC*. First, to assess transcriptional activity of *HES1*, we treated cells that were transfected with the NOTCH-responsive luciferase plasmid, *HES1*-luc, and saw a significant increase in luciferase activity upon ANGPTL2 stimulation (*Figure 5—figure supplement 1A*). Furthermore, we found significant increase in *HES1* mRNA level by qPCR during ANGPTL2-stimulation time course (*Figure 5B* blue bars). Pretreating these cells with DAPT was able to suppress these increases (*Figure 5B* red bars), implying that this is NOTCH-dependent. Similarly, we also detected significant increases in *RUNX1* and *cMYC* mRNA upon ANGPTL2 treatment that are again, NOTCH dependent (*Figure 5—figure supplement 1B*). These results indicate that ANGPTL2 can rapidly activate NOTCH as a result of induced NOTCH receptor cleavage to initiate downstream transcription of the NOTCH target genes.

NOTCH ectodomain shedding at S2 is highly regulated and thought to be key in NOTCH activation. Because the rate-limiting S2 cleavage step gives rise to the transient product, NEXT, which is then very rapidly cleaved by γ-secretase at S3 to generate the terminal fragment, NICD, we sought to determine whether ANGPTL2 can regulate S2 cleavage. To better detect NEXT accumulation, we treated ANGPTL2-stimulated human endothelial cells with the γ-secretase inhibitor, DAPT to block S3 cleavage to allow NEXT buildup. Endothelial cells have a more robust NOTCH response and because we hypothesized that *angptls* can regulate hemogenic endothelial cells in zebrafish, we felt that using human endothelial cells maybe more suitable to use in our studies. As observed in *Figure 5C*, ANGPTL2 stimulated an increase in NICD in endothelial cells (as in CD34[+] cells, *Figure 5A*) in vehicle

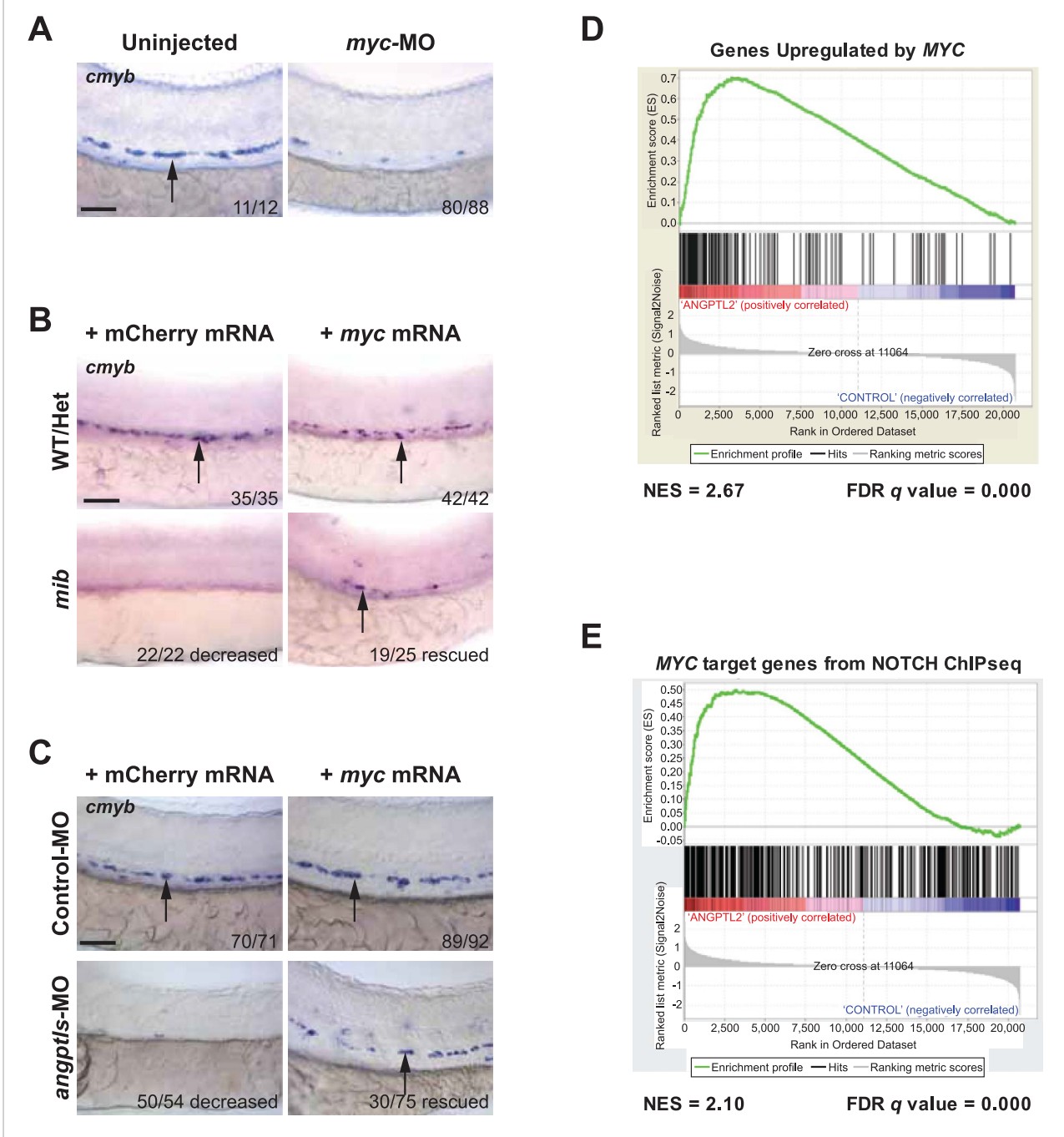

**Figure 4**. *Myc* is downstream of *angptl* and *notch* signaling. (**A**) *Myc* MO (6 ng) injected embryos had decreased *cmyb*-positive HSPCs (arrows) at 36hpf. (**B**) *Mib* embryos injected with *myc* mRNA (100 pg) had restored *cmyb*-positive HSPCs (arrows) at 36hpf compared to siblings injected with control mCherry mRNA (100 pg). (**C**) Overexpression of *myc* mRNA (100 pg) in *angptls*-MO (2 ng) morphants also had restored *cmyb*-positive HSPCs (arrows) at 36hpf compared to control siblings injected with mCherry mRNA (100 pg). Control-MO: non-targeting MO (2 ng). (**D**) Representative GSEA plot that positively correlate ANGPTL2-stimulated human CD34[+] gene expression and *MYC* targets. (**E**) GSEA plot showing significant correlation between *MYC* target genes from NOTCH ChIPseq and expression data of ANGPTL2-stimulated CD34[+] cells. NES, Normalized Enrichment Score, FDR *q* value, False Discovery Rate *q* value. Scale bars: 50 μm.

The following figure supplement is available for figure 4:

**Figure supplement 1**. NOTCH ChIP-seq in ANGPTL2-stimulated CD34[+] cells.

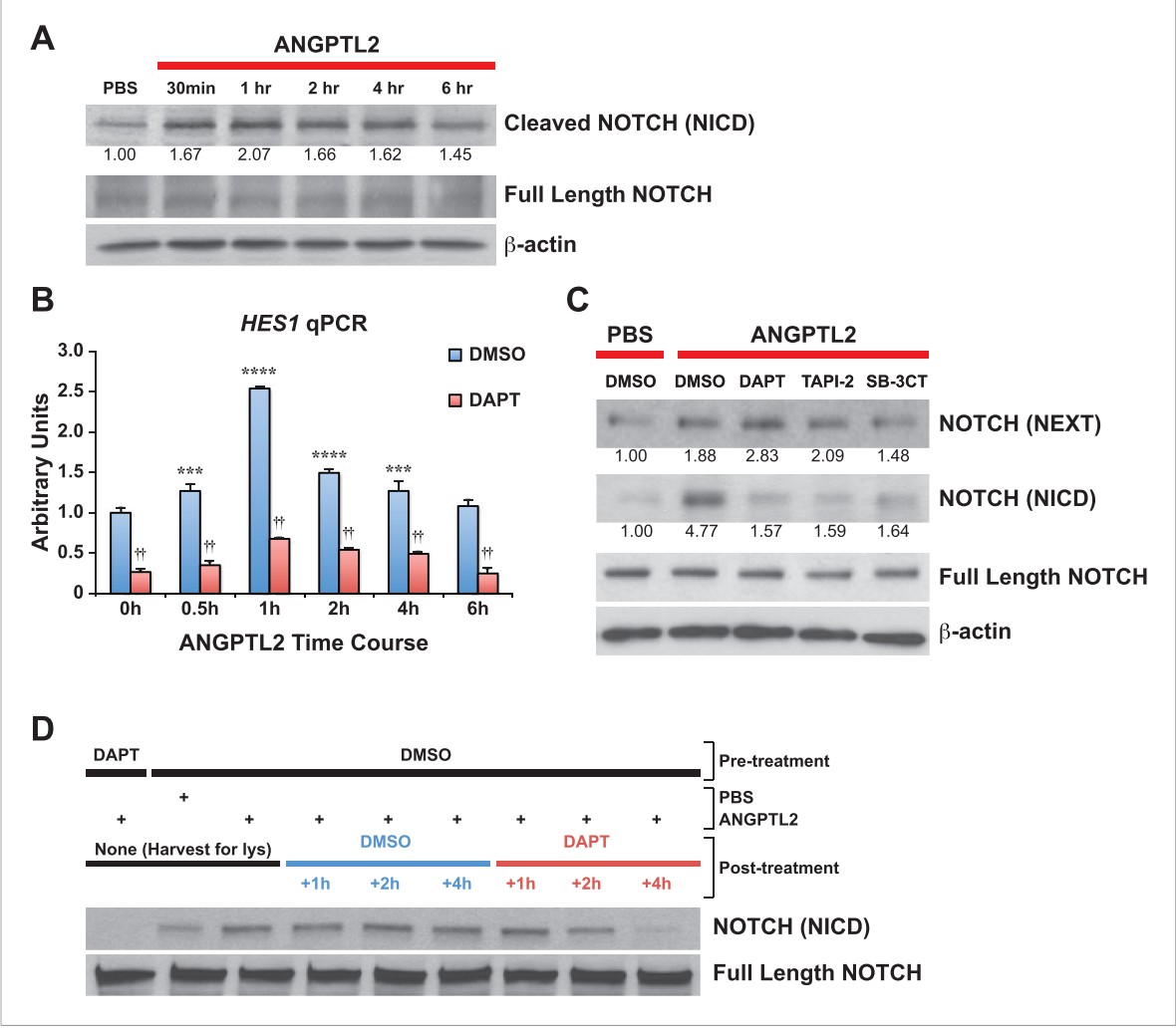

**Figure 5**. ANGPTL2 activates NOTCH by receptor cleavage. (**A**) Human CD34+ cells were stimulated with ANGPTL2 (1 µg/ml) or PBS vehicle and Western blotted for S3 cleaved (NICD) and full-length NOTCH receptor. β-actin was used as loading control. Ratios indicated below the Western blot represent densitometry of the band intensity normalized to loading control. (**B**) Human endothelial cells pre-treated for 4 hr with DMSO vehicle or DAPT (20 µM) assayed for *HES1* by qPCR. Error bars denote S.E.M., p < 0.0001, two way ANOVA; ***p < 0.001, ****p < 0.0001 compared to 0 hr and ††p < 0.0001 compared to DMSO, Bonferroni post-hoc test. (**C**) Human endothelial cells pre-treated with DMSO, DAPT (20 µM), TAPI-2 (20 µM), or SB-3CT (30 µM) prior to stimulating with ANGPTL2 (1 µg/ml) or PBS (1 hr) and Western blotted for S2 cleaved (NEXT), S3 cleaved (NICD), or full-length NOTCH. β-actin was used as loading control. Ratios indicated below the Western blots represent densitometry of the band intensity normalized to loading control. (**D**) NICD degradation experiment: post-treatment of ANGPTL2-stimulated cells (1 µg/ml for 1 hr) with DMSO or DAPT for 1, 2, or 4 hr to prevent de novo formation of NICD product, assayed by Western blotting. All experiments were repeated at least three times.

The following figure supplement is available for figure 5:

**Figure supplement 1**. ANGPTL2 activates NOTCH signaling.

(DMSO)-treated cells and this increase was blocked by DAPT. Blocking upstream of S3 at S2 using two different inhibitors of S2 cleavage (TAPI-2, a broad spectrum ADAM and TACE inhibitor; and SB-3CT, a non-competitive inhibitor for TACE) results in inhibiting NEXT and NICD formation. In control DMSO-treated cells, ANGPTL2 also stimulated the generation of NEXT fragment and this was further enhanced by treatment with DAPT, thus establishing that ANGPTL2 may regulate NOTCH activation via S2 cleavage.

The activation of Notch receptors triggers a cascade of downstream signaling and requires highly regulated mechanisms to efficiently turn off via proteasomal degradation of NICD in order to avoid deleterious effects (*Kopan and Ilagan, 2009*). Thus, to eliminate the possibility that the

ANGPTL2-mediated increase in NICD products is due to obstructed NICD degradation, we post-treated ANGPTL2-stimulated cells with DAPT to prevent further cleavage of NOTCH receptor. If the increase in NICD product following ANGPTL2 treatment is a result of NICD degradation blockade and not a result of NOTCH receptor cleavage as we postulated above, it follows that post-ANGPTL2 treatment with DAPT should inhibit any de novo NOTCH receptor cleavage resulting in sustained levels of NICD. In *Figure 5D*, we first measured NICD protein levels and found that ANGPTL2 stimulated an increase in NICD as previously observed (*Figure 5A,C*). At 1 hr post-ANGPTL2 treatment, cells were treated with either DMSO (blue bar) or DAPT (red bar). Though NICD level continued to remain high in the DMSO group, it started to decrease starting at 2 hr in the DAPT group suggesting that blocking de novo S3 cleavage was sufficient to bring ANGPTL2-stimulated NOTCH activation down to a basal state (at 4 hr of DAPT). We can conclude from this study that the rise in NICD levels induced by ANGPTL2 is a result of NOTCH receptor cleavage as opposed to interference with NICD degradation.

## ANGPTL receptor LILRB2 interacts with NOTCH

The human leukocyte immunoglobulin (Ig)-like receptor B2 (LILRB2) was identified to be the ANGPTLs receptor (*Zheng et al., 2012*). Typical of the Ig superfamily, of which includes more than 850 members, LILRB2 contains 4 Ig domains in its extracellular domain (ECD), a transmembrane domain (TM), and 3 immunoreceptor tyrosine-based inhibitory motifs (ITIMs) in its intracellular domain (ICD). Since proper compartmentalization of membrane receptors with signaling molecules is critical for coordinating and eliciting downstream cascades, we hypothesized that for ANGPTL2 to facilitate NOTCH cleavage, this interaction needs to occur proximal to NOTCH. To test this, we first transfected HEK293T cells with full-length *LILRB2* and/or *NOTCH1* and cell lysates were immunoprecipitated (IP) for NOTCH or LILRB2. Shown in *Figure 6A*, LILRB2 is bound to NOTCH and this interaction is increased upon ANGPTL2 stimulation. To assess endogenous interaction between LILRB2 and NOTCH, we performed co-IPs in human CD34+ cells and observed similarly (*Figure 6B,C*). To control for specificity of our co-IPs, we immunoprecipitated the highly abundant transmembrane protein, VE-Cadherin, in human endothelial cells and do not observe neither NOTCH nor LILRB2 association (*Figure 6—figure supplement 1A,B*).

Next, to elucidate whether this interaction occurs in *cis* or in *trans*, that is, whether LILRB2 interacts with NOTCH on the same cells or in neighboring cells, we made either N-terminal or C-terminal HA-tagged full-length *LILRB2*, *LILRB2* extracellular domain (ECD), or *LILRB2* intracellular domain (ICD) mutants, all of which still retained their TM domain for proper membrane localization (*Figure 6D* schematics). They are then singly or doubly transfected with *NOTCH* receptor into HEK293T cells prior to co-IP with NOTCH antibody and subsequently Western blotted for HA. We found that only the full-length or LILRB2 ECD can interact with NOTCH suggesting that this interaction occurs extracellularly (*Figure 6D*). Moreover, we also performed a replating experiment where singly transfected HEK293T cells (*LILRB2* ECD or ICD truncations, or *NOTCH*) were trypsinized one day after transfection, and each plate of *LILRB2* truncation cells was mixed with *NOTCH* transfected cells before being replated and allowed to grow to confluency. These cells were then processed for the same co-IP/Western blotting. We could not detect any interaction after replating (right lanes) while still being able to detect LILRB2 ECD interaction with NOTCH in double transfections (middle lanes), thus suggesting that the LILRB2 and NOTCH interaction likely occur in *cis*, that is, within the same cells extracellularly (*Figure 6—figure supplement 2*).

Now that we have established physical interaction between LILRB2 and NOTCH, we investigated whether ANGPTL2-mediated LILRB2 activation is responsible for NOTCH receptor cleavage. We made 2 stable knockdown cell lines that were lentivirally transduced for short hairpin RNA (shRNA) against *LILRB2* and performed co-IP for LILRB2. Even though we could only observe a modest knockdown of LILRB2 (*Figure 6E*, second blot, with sh-LILRB2-2 being more effective than sh-LILRB2-1), we saw that in contrast to the control scrambled shRNA transduced cells (sh-CT), which still retained the ANGPTL2-mediated NOTCH cleavage, sh-LILRB2 transduced cells had lost this response with no ANGPTL2-induced NICD formation (*Figure 6E*, third blot). Furthermore, the interaction between LILRB2 and NOTCH receptors also decreased in sh-LILRB2 cells (*Figure 6E*, first blot). Together, these studies demonstrate that ANGPTL2-induced NOTCH activation may occur through a ligand-regulated recruitment of LILRB2 to NOTCH into the same microdomain. Furthermore, this novel finding provides a plausible means by which ANGPTL2 can assist in NOTCH cleavage. The cleavage leads ultimately to an activation of *MYC* target genes that are involved in stimulating endothelial and hematopoietic cells.

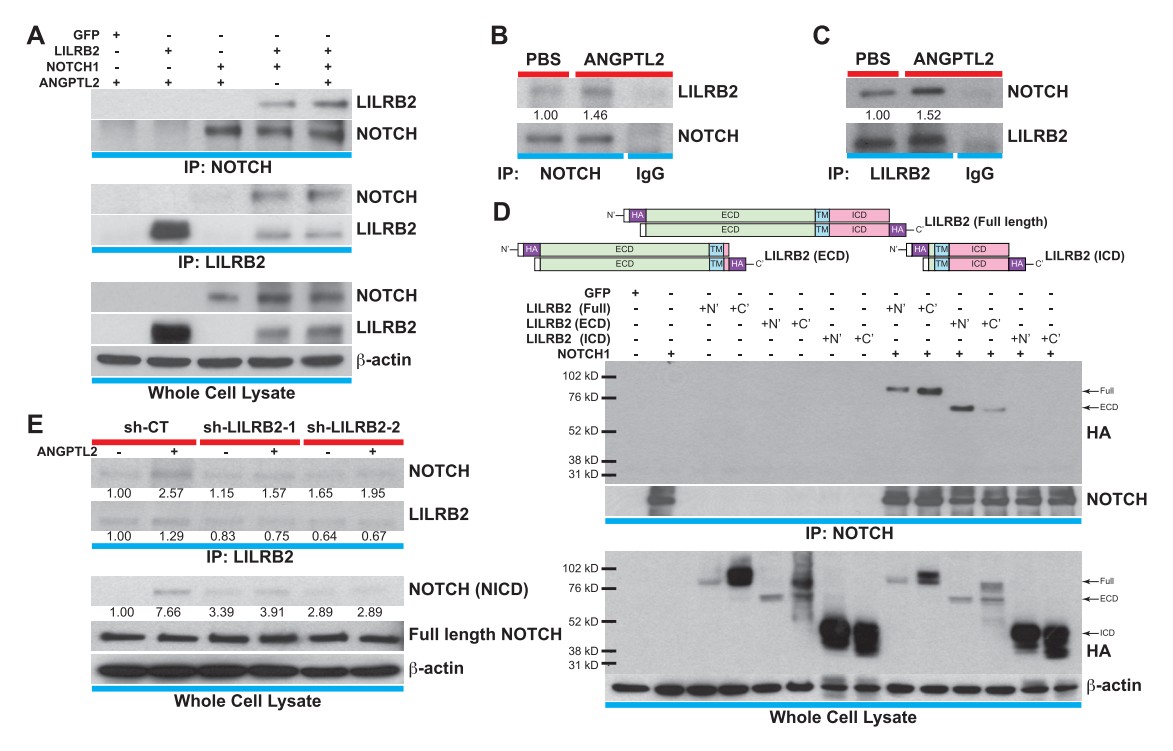

**Figure 6**. ANGPTL2 receptor, LILRB2, interacts with NOTCH in *cis*. (**A**) Cell lysates from ANGPTL2-stimulated HEK293T cells transfected with GFP control, full-length *LILRB2* and/or *NOTCH1* plasmids were IP for NOTCH1 or LILRB2. Note the interaction between both receptors and this association increased upon ANGPTL2 (1 µg/ml, 1 hr) stimulation. Whole cell lysates were used for loading control. (**B** and **C**) Cell lysates from human CD34[+] cells were co-IP for endogenous NOTCH1, endogenous LILRB2 or IgG isotype and Western blotted for associated LILRB2 or NOTCH1, respectively. (**D**) Cell lysates from *NOTCH1* and HA-tagged *LILRB2* truncation mutant (schematics) transfected cells were co-IP for NOTCH1 and Western blotted for HA. NOTCH1 interaction was only observed with full-length LILRB2 or LILRB2-ECD. (**E**) Stable lentiviral knockdown of *LILRB2* using two different sequences. Partial knockdown of LILRB2 was observed compared to the control scrambled shRNA (sh-CT). The interaction between LILRB2 and NOTCH1 was also decreased (top panels). The ANGPTL2-induced NOTCH receptor cleavage in sh-CT cells was abolished in sh-LILRB2 cells. Whole cell lysates were done for loading control. All experiments were repeated at least 3 times. Ratios indicated below the Western blots represent densitometry of the band intensity.

The following figure supplements are available for figure 6:

**Figure supplement 1**. NOTCH interacts with LILRB2 in endothelial cells.

**Figure supplement 2**. NOTCH and LILRB2 interact in *cis*.

## Discussion

Previous studies have demonstrated that *notch* is indispensable during the developmental specification of HSPCs (*Burns et al., 2005*; *Clements et al., 2011*). In our present study, we defined the essential role of *zangptl1* and *2* during developmental hematopoiesis (*Figure 1*), likely through regulation of *notch* signaling in the hemogenic endothelium. To our knowledge, this is the first time *zangptls* has been shown to activate *notch*. The current model for *notch* specification of HSPCs during development has mostly been suggested to be a cell-autonomous one (*Hadland et al., 2004*; *Burns et al., 2005*; *Robert-Moreno et al., 2005*). Our live imaging data from transient *Tg(drl:eGFP;kdrl: NICD)*-injected embryos resulting in significantly more budding HSPCs in the AGM of DA corroborate with this. More recently, studies have revealed an additional and separate role for somite-derived *notch* signaling during HSPC specification that occurs much earlier than previously thought (*Clements et al., 2011*; *Kobayashi et al., 2014*). Based on our genetic data in zebrafish in which we observed a tight genetic interaction between *zangptls* and *notch* (*Figure 2*), we believe that *zangptls* exert their effects on definitive hematopoiesis during the vascular specifications of the axial vessels including the

DA, when it coincides with the highest endogenous expression of *zangptl2* (*Figure 1—figure supplement 1B*). Though it remains to be determined, information on the spatial/temporal expression pattern of the yet unidentified *zangptl* receptor would solidify this premise. There are receptors in zebrafish with similar domains as the mammalian receptor, but there is no obvious orthologue. Nevertheless, our current proposed model (*Figure 7A*) extrapolated from our genetic data places *zangptl2* downstream of *mib* as *Tg(hsp70:zangptl2)* crossed into *mib* can rescue its hematopoietic defect (*Figure 2C*). *Zangptl2* acts upstream of *notch* activation, that is, generation of *NICD*, as forced expression of *NICD* from *Tg(hsp70:Gal4;UAS:NICD)* can rescue *angptls*-MO morphant phenotype (*Figure 2B*). Interestingly, loss of *notch* signaling in *angptls*-MO-injected *notch* reporter embryos (*Figure 2A*) indicates that *zangptls* are required for *notch* signaling but also raises questions as to whether *notch* ligands needed to be present for *zangptls* to regulate *notch* activation. One of the *notch* ligands, *deltaC,* has considerable overlapping expression both spatially and temporally with *notch* in the developing DA, suggesting that it may play a role during the formation of the hemogenic endothelium and the subsequent generation of HSPCs. *DeltaC* expression, normally absent in the DA of *mib* was similarly rescued in *Tg(hsp70:zangptl2);mib* (data not shown). This restoration of *notch* ligand in the *mib* by *zangptl2* overexpression was intriguing, and it remains to be established whether this restored *notch* ligand is functional in the absence of *mib*.

To examine potential downstream signaling from *angptls* and *notch,* we looked at the previously identified target of *notch, myc.* The relationship between *NOTCH* and *MYC* has been best studied in T-ALL cells whereby *MYC* transduces growth and survival signals to many *NOTCH*-dependent T-ALL cell lines and forced ectopic expression of *MYC* can restore the leukemogenic signals for these cells in instances when *NOTCH* is inhibited (*Palomero et al., 2006*; *Weng et al., 2006*; *Chan et al., 2007*). However, *myc* has not been studied in zebrafish hematopoiesis. We found that not only is *myc* required during HSPC formation in the AGM (*Figure 4A*), overexpression of exogenous *myc* in embryos lacking *zangptls* (*angptls*-MO) or *notch* (*mib*) signaling could similarly rescue AGM defects (*Figure 4C,B*). This puts *myc* downstream of *zangptls* and *notch* signaling. Through independent gene expression analysis, we found *MYC* to be highly activated upon ANGPTL2 stimulation in human CD34$^+$ progenitor cells (*Figure 4D*). Furthermore, we ran GSEA comparing our ANGPTL2-induced gene expression signature to the set of *MYC* target genes whose regulatory elements were bound by NOTCH from our ChIP-seq data and found significant overlap (*Figure 4E*). This implies that the set of *MYC* targets downstream of ANGPTL2 and NOTCH signaling is highly similar, supporting our hypothesis that ANGPTL may signal partially through NOTCH.

To test this hypothesis, we utilized cultured mammalian cells to directly investigate the mechanism by which ANGPTL2 may act on NOTCH. We first discovered that purified ANGPTL2 ligand added to cells can rapidly induce NOTCH activation, measurable using three different assays: a luciferase reporter assay (*Figure 5—figure supplement 1A*), qPCR of known NOTCH target genes (*Figure 5B* and *Figure 5—figure supplement 1B*), and Western blotting for the NOTCH cleavage product NICD (*Figure 5A*). Of particular note, when we used different drugs that block certain cleavage events, we found that ANGPTL2-stimulated S2 cleavage of NOTCH (*Figure 5C*). Interestingly, this step occurs at the cell surface when NOTCH engages with its ligands. Although the precise mechanism in which ANGPTL2 mediates NOTCH cleavage is subject to further investigation, our assumption that this level of regulation must occur in close proximity led to our surprising discovery that human ANGPTL2 receptor, LILRB2, can interact with NOTCH (*Figure 6A–D* and *Figure 6—figure supplement 1,2*). These preliminary results from co-IPs certainly suggest that ANGPTL signaling occurs in very close proximity to NOTCH. Such co-IP experiments do not definitively signify that there is direct physical interaction and do not address the stoichiometry of this potential interaction. Nevertheless, ANGPTL2-mediated NOTCH cleavage appears to be dependent on LILRB2 signaling, as sh-LILRB2 knockdown resulted in diminished NICD generation (*Figure 6E*). This demonstrates a functional interaction. A more thorough mapping of the interactions between these two surface receptors will shed light on the mechanism by which ANGPTL2 can regulate NOTCH activation. An attractive speculation as to why this interaction may occur would be to potentially facilitate or even poise the NOTCH receptor for more efficient cleavage, as depicted in the model proposed in *Figure 7B*. This also raises an intriguing possibility of whether LILRB2 can physically alter NOTCH interaction with its ligand or whether ANGPTL2 signaling can influence NOTCH ligand availability. Indeed several studies have suggested that Notch receptors oligomerize at the cell surface at sites of DSL ligand contact from neighboring cells (*Luty et al., 2007*; *Nichols et al., 2007*). The

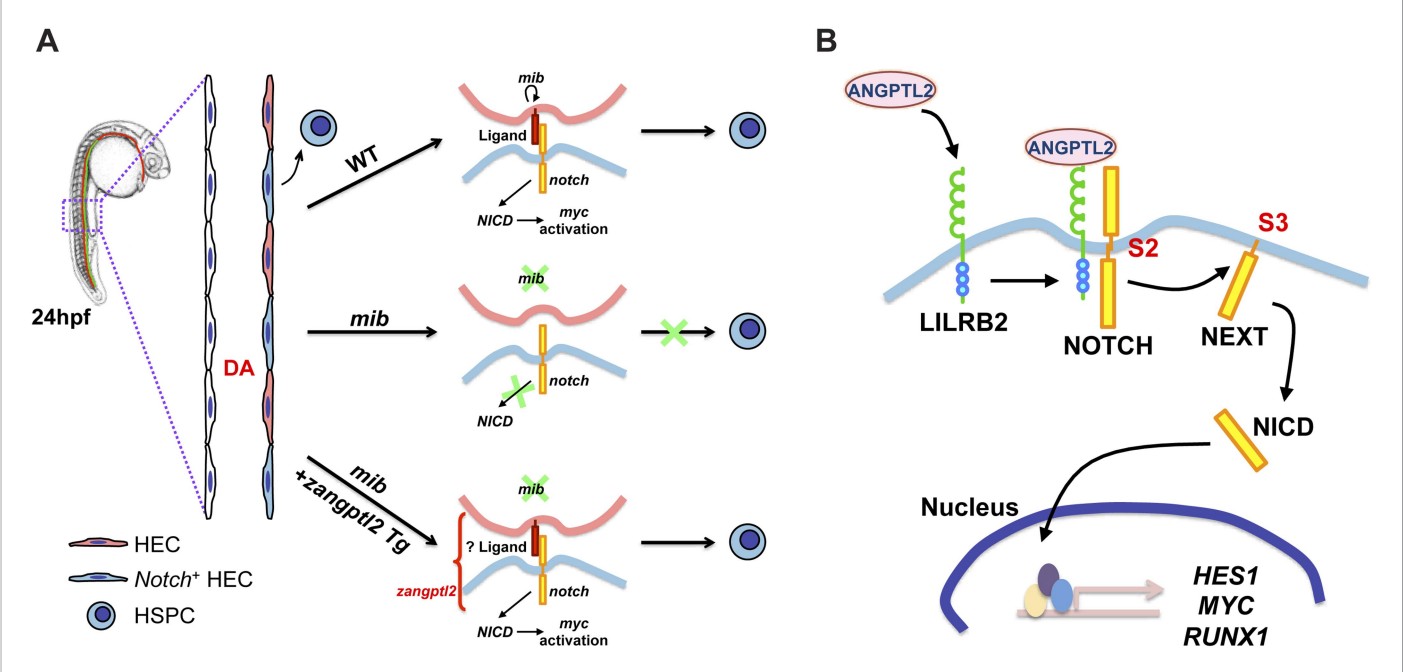

**Figure 7**. Proposed models for ANGPTL-mediated NOTCH signaling. (**A**) *Notch* signaling in zebrafish is believed to be important for HSPC specification in the hemogenic endothelium on the ventral side of the developing dorsal aorta (DA). A simplified view of canonical *notch* signaling starts with *mib* endocytic processing of *notch* ligands to potentiate their ability to activate *notch*, depicted in the wild-type (WT) model. *Notch* receptors interact with ligands in the neighboring cell and this leads to subsequent cleavage of *notch* to release *NICD* to translocate into the nucleus and initiate transcription of target gene like *myc*. In the *mib* mutant however, the lack of signal from *notch* ligands prevents further downstream signaling from the receptor and HSPCs are not formed. From our genetic interaction studies, we believe that *zangptl2* is downstream of *mib* (**Figure 2C**) but upstream of *NICD* generation (**Figure 2B**). (**B**) Using cultured cells to further explore the mechanism by which ANGPTL2 can regulate NOTCH signaling, we found that ANGPTL2 can stimulate NOTCH receptor cleavage at S2 and S3, leading to transcription of target genes. This was dependent on the ANGPTL receptor, LILRB2, as ANGPTL2 can induce recruitment of LILRB2 to NOTCH.

recruitment of LILRB2 to NOTCH may facilitate this clustering process although the stoichiometry of LILRB2 to NOTCH remains to be established.

Various alternative non-DSL type ligands have been identified to activate NOTCH, such as the adhesion molecule F3/Contactin (*Hu et al., 2003*), EGF-repeat factor DNER (*Eiraku et al., 2005*), EGF-like domain 7 (*Schmidt et al., 2009*), and Microfibrillar associated glycoproteins (MAGP) 1 and 2 (*Miyamoto et al., 2006*). Interestingly, a related family member to the latter, MFAP4 was found to be capable of expanding mouse HSPCs *ex vivo* along with ANGPTLs 2 and 3 (*Zhang et al., 2006*). Our study does not propose ANGPTL2/LILRB2 as a non-canonical NOTCH ligand but offers an additional layer of regulation on NOTCH activation which may not be previously appreciated.

Previous report on the discovery of ANGPTL receptor, LILRB2, suggested downstream activation of CAMK (calcium/calmodulin-dependent protein kinase)-2 and -4 and subsequent recruitment of SHP-2 (a Src homology 2 domain containing non-transmembrane protein tyrosine phosphatase) in freshly isolated mouse spleen cells (*Zheng et al., 2012*). CAMK2 has been shown to activate NOTCH in human prostate cancer cells (*Mamaeva et al., 2009*) whereas SHP-2 has also been shown to genetically interact with *notch* signaling in *Drosophila* (*Oishi et al., 2006*). Although the answer as to whether CAMKs and SHP-2 may directly or indirectly regulate NOTCH is still yet to be determined, this suggests that LILRB2 signaling may extend beyond the models presented in our studies.

The ability of ANGPTLs to regulate the local events at which NOTCH cleavage occurs creates new opportunity for therapeutic intervention. The role of Notch during HSPC renewal at homeostasis is controversial (*Bigas et al., 2010*). Recent studies demonstrate that endothelial-derived *Notch* signal can stimulate murine cKit[+]Sca1[+]Lin[−] cells expansion (*Butler et al., 2010*). Delta1[ext−IgG], in which the engineered activating NOTCH ligand is tethered to the Fc domain of human IgG1, was developed for ex vivo expansion of human CD34[+] cord blood (CB) progenitors (*Delaney et al., 2005*). More recently,

Delta1[ext–IgG]-expanded HSPCs showed great clinical promise in rapid hematopoietic engraftment with shortened time to neutrophil recovery (*Delaney et al., 2010*). In support of an interaction between the *angptls* and *notch*, preliminary analysis of human CD34[+] CB progenitors treated with ANGPTL2 or Delta[ext–IgG] had comparable expansion in CD34[+]/CD90(Thy-1)[lo] populations. Furthermore, these expansions were effectively abrogated when cells were treated with a combination of NOTCH1 and NOTCH2 blocking antibodies, suggesting that they were at least partially NOTCH mediated. This supports that the ANGPTLs and NOTCH can interact in another cell type such as human cord blood CD34[+] cells. In conclusion, our studies of ANGPTL-mediated activation of NOTCH presented here have uncovered an important genetic, physical, and functional interaction of these two signaling pathways that are critical for hematopoiesis.

# Materials and methods

## Zebrafish strains and heatshock experiment

Zebrafish were maintained in accordance to Boston Children's Hospital Animal Research Guidelines. The following mutants and transgenic zebrafish were used in the study: *mindbomb*[ta56b] (*Jiang et al., 1996*), *Tg(UAS:NICD)* (*Scheer et al., 2001*), *Tg(hsp:70:Gal4)* (*Scheer et al., 2001*), *Tg(Tp1bglob: eGFP)*[um14] (*Parsons et al., 2009*), *Tg(kdrl:Hras-mCherry)* (*Chi et al., 2008*), or *Tg(Runx1+23:NLS-mCherry)* (*Tamplin et al., 2015*) crossed into the casper (*White et al., 2008*) background and *Tg (hsp70:zangptl2;cmlc2:DsRed2)*. The latter was generated by co-injecting linearized plasmids: one with a 1.5 kb heatshock protein 70 promoter (*pzhsp70/4prom*) driving full-length *D. rerio angptl2* (provided by Y Kubota and T Suda) and the other containing the 5.1 kb cardiac myosin light chain promoter 2 (*Rottbauer et al., 2002*) driving nuclear DsRed2. A stably integrated transgenic line containing both transgenes was maintained, and transgenic embryos were identified with DsRed2-positive heart. To induce heatshock overexpression of *zangptl2*, embryos at the 10–12 ss were immersed in E3 fish water and heated to 38°C for 30 min to 1 hr. Similarly, to induce heatshock overexpression of NICD, embryos from *Tg(UAS:NICD)* and *Tg(hsp:70:Gal4)* crosses were heatshocked at 39°C for 20 min. Embryos were quickly washed with room temperature E3 to stop heatshock and allowed to develop normally until the appropriate stage before fixation with 4% paraformaldehyde and processed for WISH. Embryos from at least three separate breeding clutches were scored for each experiment and tabulated together for the observed phenotype. The results are represented as a single ratio of those depicted in each panel to the total number of embryos scored. In instances where a genetic cross were done, we scored within each breeding clutch, the different genotypes first, then the observed phenotype. The denominator in the ratios indicated in each panel represents the total number of embryos scored within that genotype. After the WISH pictures were taken, all transgenic and mutant embryos had their genotype confirmed by PCR. Genotyping PCR primers for *mib*-F: GGTGTGTCTGGATCGTCTGAAGAAC, *mib*-R: GATGGATGTGGTAACACTGATGACTC, *UAS:NICD*-F: CATCGCGTCTCAGCCTCAC, *UAS:NICD*-R: CGGAATCGTTTATTGGTGTCG, *hsp:70:Gal4*-F: GCAAT-GAACAGACGGGCATTTAC, *hsp:70:Gal4*-R: CTTCAGACACTTGGCGCACTTCGG, *hsp:70:zangptl2*-F: CAGAGAAACTCAACCGAAGAGAAGCGAC, *hsp:70:zangptl2*-R: GCTCCTGTAACCTTCTGCTGGGGTA, *cmlc2:DsRed2*-F: TGTATTTAGGAGGCTCTGGGTGTC, and *cmlc2:DsRed2*-R: CTTCTTGTAGTCGGG-GATGTCG. Each experiment was repeated at least twice.

## WISH, morpholinos, and mRNA microinjections

WISH was performed as described (*Thisse and Thisse, 2008*). Previously published morpholinos (Gene Tools, Philomath, OR) used to knockdown *angptl1* and *angptl2* were targeted to the start codons or 5′ UTR (*Kubota et al., 2005*). Additional morpholinos used to confirm knockdown was generated as splicing morpholinos with the sequences: *angptl1*: 5′-CCTGTGGAAAATGCAGAGAAATGCA-3′ and *angptl2*: 5′-GAGGTTTTTCTTGTGGCTCACCTTA-3′. Control non-targeting morpholino sequence was 5′-CCTCTTACCTCAGTTACAATTTATA-3′. *Myc* morpholino sequence was 5′-GTGGTAAAAGCTGAATGAACACTGA-3′. mRNA for *D. rerio myc* (provided by A Gutierrez) was synthesized using the mMESSAGE mMACHINE kit (Life Technologies, Grand Island, NY) per manufacturer's protocol. For every experiment, equal amounts of morpholinos and/or mRNAs are always controlled for by injecting the non-targeting control morpholinos and/or mCherry mRNA into sibling embryos at 1 cell stage as previously described (*Burns et al., 2005*).

## qPCR and primer sequences

RNA was extracted from whole embryos (average of 50 embryos per condition, three clutches of embryos per condition) or cells (average of $10^6$ cells, triplicates per experiment) using TRIzol (Life Technologies, Grand Island, NY) or RNeasy miniprep (Qiagen, Germantown, MD) respectively, followed by DNaseI (Qiagen, Germantown, MD) digestion and RNeasy (Qiagen, Germantown, MD) cleanup. cDNA was synthesized using SuperScript VILO cDNA Synthesis Kit (Life Technologies, Grand Island, NY). Between 1 and 5 ng of cDNA was used per qPCR reaction with 200 µM primers using the iQ SYBR Green Supermix (Bio-Rad, Hercules, CA) on CFX384 Real-Time PCR Detection System (Bio-Rad, Hercules, CA). qPCR primers used: *zangptl2*-F: TCAGAGTGGGCCGTTATCATGGAA, *zangptl2*-R: TGATAACGACTGC GGTAATGCCCT, *HES1*-F: ATAGCTCGCGGCATTCCAAGC, *HES1*-R: CCAGCACACTTGGGTCTG TGCT, *RUNX1*-F: AGGAAGACACAGCACCCTGGA, *RUNX1*-R: ACGTGCATTCTGAGGGCTGTCA, *MYC*-F: CGACTCTGAGGAGGAACAAG, *MYC*-R: GTGCGCACCTCGGTATTAAC, *GAPDH*-F: CCTGCA CCACCAACTGCTTA, *GAPDH*-R: CCATCACGCCACAGTTTCC (*GAPDH* used as normalizing gene).

## Time-lapse live imaging of zebrafish embryos

The 6.4-kb regulatory region upstream of *drl* was cloned into the backbone of the Tol2 destination vector, #394 pDESTTol2pA2 (from the Tol2kit [Kwan et al., 2007]) driving eGFP. The resultant vector was then recombined using the Multisite Gateway technology (Life Technologies, Grand Island, NY) using 5′ entry clones containing the *kdrl* promoter (p5E-kdrl, provided by C.B. Chien) or the Runx1+23 enhancer (p5E-Runx1+23 provided by OJ Tamplin); middle entry clones containing eGFP (#383 pME-eGFP, Tol2kit) or constitutively active *NOTCH1* (pME-NICD); and 3′ entry clones containing SV40 late pA (#302 p3E-pA, Tol2kit). All vectors were sequence verified prior to injection into 1 cell stage *Tg (kdrl:Hras-mCherry)* or *Tg(Runx1+23:NLS-mCherry)* embryos at 25 ng/embryo. 15 pg of Tol2 (transposase enzyme) mRNA was co-injected per embryo. Dechorionated and staged embryos were imaged beginning at 28hpf and 72hpf for the AGM and CHT, respectively. To control for variability in injection, embryos were pre-screened for a similar degree of mosaicism. Selected embryos were mounted in glass bottom 6-well plates (MaTek, No. 1.5 cover glass) in 1% LMP agarose in E3 embryo medium containing 0.02% Tricaine. Time-lapse imaging was performed using a Yokogawa CSU-X spinning disk and Andor iXon EM-CCD cameras mounted on an inverted Nikon Eclipse Ti microscope (Andor Technology, South Windsor, CT). 488 nm and 561 nm lasers were used to image GFP and mCherry, respectively. Control eGFP and NICD-injected embryos were imaged together in separate wells of the same 6-well plate on the same day to control for imaging variability. All imaging was done within a 28.5°C humidified chamber. Confocal z-stacks were collected every 10 min using a motorized stage piezo, acquiring 5 µm z-steps across 120 µm, tiling three fields of view for the AGM, or 5 µm z-steps across 60 µm, tiling five fields of view for the CHT. 3D time-lapse video files were rendered as maximum image projections using Imaris (Bitplane, South Windsor, CT).

## Cell culture, ANGPTL2 protein expression, and purification

Human endothelial cells (either human umbilical vein endothelial cells [primary or EA.hy926] or human aortic endothelial cells) or the leukemia cell line K562 were cultured per manufacturer's protocol. Human hematopoietic CD34$^+$ progenitor cells isolated from peripheral blood of GCSF-mobilized healthy volunteers were obtained from the Fred Hutchinson Cancer Research Center. These cells were expanded for 6 days in StemSpan SFEM (Stem Cell Technology, Canada) supplemented with CC100 cytokine mix and 2% Pen/Strep. We have previously found that these cells will remain at least 75–80% CD34$^+$ and greater than 90% CD38$^-$ after this cultured period (Trompouki et al., 2011). Though they do not truly represent HSCs after culture, they nevertheless retain the capacity to differentiate and thus are utilized as hematopoietic progenitors in our studies. Prior to stimulation with purified FLAG-tagged ANGPTL2 ligand, cells were either serum starved (endothelial or K562 cells) or cytokine starved (CD34$^+$ cells) for 2–16 hr at which time they were drug treated or vehicle treated (DMSO). FLAG-tagged murine ANGPTL2 (vector provided by Y Oike) was expressed in HEK293T cells for 72 hr and cell supernatant was purified as previously described (Zhang et al., 2006). Briefly, cell supernatant was centrifuged to remove cell debris and supplemented to a final concentration of 150 mM NaCl. Complete Protease Inhibitor Cocktail (Roche, Indianapolis, IN) was added at 1 tablet/50 ml. Cell supernatant was pre-cleared using unconjugated Sepharose beads prior to incubation with the anti-FLAG M2 affinity Gel (Sigma, St. Louis, MO) overnight at 4°C. Gel column was washed with TBS

8–10 times and eluted using FLAG peptide at 150 µg/ml, dissolved in TBS. Eluent was dialyzed against 4 changes of PBS and analyzed on SDS-PAGE to determine protein concentration.

## Luciferase reporter assays

The Notch firefly luciferase reporter plasmid pGL2-Hes1 reporter (Hes1-luc, provided by JC Aster) and pRL-TK (Promega, Madison, WI) encoding the *Renilla* luciferase plasmid were co-transfected into K562 cells using AMAXA nucleofector according to the manufacturer protocols (Lonza, Hopkinton, MA). Cells were lysed and assayed using the Dual-Luciferase Reporter Assay System (Promega, Madison, WI). Firefly luciferase activity was normalized to *Renilla* luciferase activity and plotted as a ratio of the two.

## LILRB2 truncation mutants

Full-length human LILRB2 and intracellular (ICD) truncation mutant were obtained from CC Zhang. Full-length LILRB2 contains four extracellular (ECD) IgG domains, a transmembrane domain (TM) and a short ICD. The LILRB2 ECD truncation mutant includes the entire ECD of LILRB2, the transmembrane domain and the first six amino acids from the ICD. The LILRB2 ICD truncation mutant includes 41 amino acids from the end of ECD, TM and the entire ICD. The HA-tag (TACCCATACGATGTTCCA GATTACGCT) followed by a linker sequence (GGAGGCTCAGGGGGGTTCC) was cloned 5′ to the ATG of full-length LILRB2 or LILRB2 ECD to make the N-terminal-tagged mutants. Likewise, the linker sequence followed by the HA tag was cloned 3′ to the last amino acid before STOP of full-length LILRB2 and LILRB2 ECD to make the C-terminal-tagged mutants. To generate N- or C-terminal HA-tagged LILRB2-ICD, we performed site-directed mutagenesis to insert the HA and linker sequences (Q5 Site-Directed Mutagenesis Kit, NEB, Ipswich, MA).

## LILRB2 shRNA infection and virus production

Lentiviral shRNA in the pLKO.1-puromycin vector were obtained from the Sigma Mission shRNA library. Two lentiviral shRNA constructs targeted against human LILRB2 were obtained with the following sequences: sh-LILRB2-1 (TRCN0000416153: CCGGGAAGTAAGAATGTGCTT TAAACTCGAGTTTAAAGCACATTCTTACTTCTTTTTTG) and sh-LILRB2-2 (TRCN0000416926: CCGGTGACGTTGGCTTTCGTATAAGCTCGAGCTTATACGAAAGCCAACGTCATTTTTTG). An shRNA with a scrambled sequence was used as a non-targeting control (sh-CT). HEK293 cells were cultured in Dulbecco's modified Eagle's Medium supplemented with 10% fetal bovine serum and 2% penicillin-streptomycin (Life Technologies, Grand Island, NY). They were transfected at 80% confluence using branched polyethylenimine (Sigma, St. Louis, MO) with psPAX2 and VSV-G lentiviral plasmids as well as the relevant shRNA plasmid. Medium was refreshed 16–24 hr after transfection. Medium containing viral particles was collected at 48 and 72 hr after transfection and concentrated by ultracentrifugation. Stable EA.hy926 cells were infected with each virus with 8 µg/ml protamine sulfate (Sigma, St. Louis, MO) overnight at 37°C and selected for 2 weeks in 1 µg/ml Puromycin (Sigma, St. Louis, MO).

## Immunoprecipitation and Western blotting

Subsequent to cells stimulated with purified ANGPTL2, cells were washed with ice-cold PBS prior to lysing in RIPA buffer (Life Technologies, Grand Island, NY) supplemented with 20 mM NaF, 1 mM $Na_4P_2O_7$, 0.3 mg/ml Pefabloc SC (Roche, Indianapolis, IN), Complete Protease Inhibitor Cocktail (Roche, Indianapolis, IN), and 1 mM $Na_3VO_4$. Normalized protein lysates were diluted to 1 mg/ml prior to pre-clearing using Protein G-conjugated Dynal beads (Life Technologies, Grand Island, NY). Pre-cleared lysates were immunoprecipitated with the appropriate antibodies overnight at 4°C. Protein G-conjugated Dynal beads were used to pull down immunocomplexes and washed with IP buffer (Life Technologies, Grand Island, NY), supplemented with the above mentioned protease/phosphatase inhibitors, three times prior to eluting in SDS sample buffer and loading onto SDS-PAGE for Western blotting. Antibodies used to perform co-IPs and Western blotting are Notch1 (Santa Cruz, Dallas, TX, sc-6014R or Cell Signaling, Danvers, MA, #3608), Cleaved Notch Val1744 (NICD, Cell Signaling, Danvers, MA, #4147), Cleaved Notch (NEXT, provided by S Blacklow), LILRB2 (Santa Cruz, Dallas, TX, sc-33454), β-actin (Sigma, St. Louis, MO, #A2228), HA (Sigma, St. Louis, MO, #H9658), and VE-Cadherin (Santa Cruz, Dallas, TX, sc-6458). All co-IPs and Westerns were repeated at least three times with similar results. The densitometries of Western blots are measured using ImageJ v1.48 and indicated below the blots as a ratio of band intensity normalized to the loading control β-actin or the protein used for IP.

## Microarray analysis

Samples for microarray analysis were obtained by isolating RNA from three biological triplicates each of human CD34[+] progenitor cells that have been stimulated with ANGPTL2 (1 µg/ml) or PBS vehicle using TRIzol (Life Technologies, Grand Island, NY) extraction followed by DNaseI (Qiagen, Germantown, MD) digestion and RNeasy (Qiagen, Germantown, MD) cleanup. RNA was prepared for hybridization to the Affymetrix Human Exon 1.0 ST Arrays. Raw CEL files were analyzed using the Genepattern software suite (Broad Institute, Cambridge, MA). Raw CEL files were converted into GCT format using the ExpressionFileCreator Module, with RMA and quantile normalization. The GCT file was then used in the PreprocessDataSet module, where threshold and ceiling values were set (floor 20 ceiling 20,000, fold = 1.5, delta = 1), and any value lower/higher than the threshold/ceiling were reset to the threshold/ceiling value, and data were processed to discard any invariant genes. Comparisons between groups were performed using the ComparativeMarkerSelection modules, using two-sided t-test with Benjamini-Hochberg multiple hypothesis testing. The data for these microarrays are available on NCBI Gene Expression Omnibus database under GEO accession number GES51652.

## GSEA analysis

The data generated from the Affymetrix arrays were used to query the Broad Molecular Signature Database. The GCT files generated by Genepattern were used as input to GSEA (version 2, Broad Institute) and queried against the c2 curated gene sets, c3 motif gene sets, and c5 GO gene sets. GSEA parameters used for this analysis: scoring_scheme=weighted, metric=Signa2Noise, permute_type= gene_set, permutations=1000. Genesets with a FDR<0.25 were considered significant, and the top 20 sets (all with FDR = 0.00) were used for Leading Edge analysis. The complete list of GSEA results can be found in *Supplementary File 1*.

## ChIP-seq sample preparation, Polony generation, and sequencing

Samples were prepared as previously described (*Trompouki et al., 2011*). Briefly, 10[8] of the ANGPTL2-stimulated human peripheral blood CD34[+] progenitors were cross-linked with fresh formaldehyde, quenched with Glycine and lysed. The nuclear extracts were sonicated (Bioruptor) for 36 cycles of 30-s followed by 1-min rest prior immunoprecipitation with 100 µg of NOTCH antibody (*Wang et al., 2011*) and 100 µg of Protein G Dynal beads overnight at 4°C. 50 µl of sonicated lysates prior to antibody addition was reserved as input sample. Beads are washed, eluted, and reverse cross-linked before RNase and Proteinase K treatment and DNA extracted by Phenol/Chloroform. ChIP or input DNA overhangs were blunted and purified using PCR purification kit (Qiagen, Germantown, MD). Single A in the 3′ end was added and purified with the MinElute PCR purification kit (Qiagen, Germantown, MD). Illumina adaptor oligos were added, followed by PCR purification again. The samples were amplified by PCR (18 cycles) to add linker sequence to the fragments to prepare for annealing to the Genome Analyzer flow-cell. Amplified samples were separated on a 2% agarose gel, and products between 150 and 350 bp were excised and gel extracted using Gel Extraction kit (Qiagen, Germantown, MD). Polony generation and sequencing were done as previously described (*Trompouki et al., 2011*).

## ChIP-seq data analysis

Sequences obtained from the Illumina/Solexa sequencer were processed through the bundled Solexa image extraction pipeline as previously described (*Trompouki et al., 2011*). Briefly, sequences were aligned using ELAND software to NCBI Build 36 (UCSC hg18) of the human genome. Only sequences that mapped uniquely to the genome with zero or one mismatch were used for our analysis. Analysis methods were based on previously published methods (*Johnson et al., 2007*; *Guenther et al., 2008*; *Marson et al., 2008*). Each read was extended 200 bp toward the interior of the sequence fragment, based on the strand of the alignment. Across the genome the number of ChIP-seq reads was tabulated in 10 bp bins, and the genomic bins that contained statistically significant ChIP-seq enrichment were identified by comparison to a Poissonian background model. Assuming background reads are spread randomly throughout the genome, the probability of observing a given number of reads in a 1-kb window can be modeled as a Poisson process in which the expectation can be estimated as the number of mapped reads multiplied by the number of bins into which each read maps, divided by the total number of bins available. Enriched bins within 200 bp of one another were

combined into regions. The Poissonian background model assumes a random distribution of background reads. However, significant deviations from this expectation have been observed. Some of these non random events can be detected as sites of apparent enrichment in negative control DNA samples creating false positives. To remove these false-positive regions, negative control DNA from whole-cell extract (Input DNA sample) was sequenced. Enriched bins and enriched regions were defined as having greater than fivefold density in the experimental sample compared with the control sample when normalized to the total number of reads in each data set. This served to filter out genomic regions that are biased to having a greater than expected background density of ChIP-seq reads. Enriched regions within 5 kb upstream or downstream of the body of the gene were called bound. Additionally, data files that contain genome browser tracks showing genome-wide ChIP-seq density and enriched regions for all experiments are available on NCBI Gene Expression Omnibus database under GEO accession number GES63010.

## Motif analysis, GREAT analysis, and Ingenuity Pathway analysis

Non-repeat sequences from NOTCH ChIP-seq were uploaded onto MEME-ChIP (http://meme. nbcr.net/meme/tools/meme-chip) for motif analysis. Enriched regions from the NOTCH ChIP-seq were imported into Genomic Regions Enrichment Annotations Tool (GREAT) (*McLean et al., 2010*) and genomic regions that are associated with putative genes were used to generate terms that include gene ontology, phenotype data and human disease, pathway data, gene expression, regulatory motifs, and gene families. Each term is determined by the region–gene association settings and significance is computed from the enriched regions of the NOTCH ChIP-seq falling in the regulatory domains of genes involved in a particular function compared to random. Both the enriched genes from the NOTCH ChIP-seq and the gene lists from the microarrays were imported into Ingenuity Pathway Analysis (IPA). The functional analysis identified the biological functions and/or disease states that are most significant to the data set. Of note, the Upstream Regulator analysis was used to predict which molecules (in its active or repressed state) are likely to give rise to the observed expression data. *MYC* was identified as the top active molecule.

## Acknowledgements

We would like to thank Y Kubota, Y Oike, T Suda, S Blacklow, H Wang, Y L Wong, K Vasudevan J Ganis, J Lahvic, I Shestopalov, and A Gutierrez for providing reagents; C Blobel, S Blacklow, and TA Springer for helpful discussion and critical comments; Y Zhou for bioinformatics discussions; and N Lawson for providing the Tg*(Tp1bglob:eGFP)*$^{um14}$. The microarray expression experiments were processed by the Microarray Core Facility of the Molecular Genetics Core Facility at Children's Hospital Boston supported by NIH P50-NS40828 and NIH P30-HD18655. This work was supported by NIH 5R01HL048801-21, NIH 5P30DK49216-19, NIH R24DK092760-02, NIH 5U01HL10001-05, NIH 5PO1HL32262-32, NIH 5R01DK53298 and HHMI (to LIZ); Canadian Institutes of Health Research Fellowship and the American Heart Association Postdoctoral Fellowship, 11POST4920031 (to MIL); Hyundai Hope On Wheels Grant (to AD); NIH 1U01HL100395-06 and Leukemia & Lymphoma Society TRP 6407-13 (to IDB); NIH K08AR055368, Melanoma Research Alliance Young Investigator Award, and ASCO/AACR Young Investigator Award (to RMW), NIH 1R01CA172268 (to CCZ). L.I.Z. is a founder and stockholder of Fate, Inc. and Scholar Rock, and a scientific advisor for Stemgent.

## Additional information

### Competing interests

LIZ: I am a founder and stockholder of Fate, Inc. and Scholar Rock, and a scientific advisor for Stemgent. The other authors declare that no competing interests exist.

### Funding

| Funder | Grant reference | Author |
|---|---|---|
| National Institutes of Health (NIH) | 5R01HL048801-21 | Leonard I Zon |
| National Institutes of Health (NIH) | 5P30DK49216-19 | Leonard I Zon |

| Funder | Grant reference | Author |
|---|---|---|
| National Institutes of Health (NIH) | R24DK092760-02 | Leonard I Zon |
| National Institutes of Health (NIH) | 5U01HL10001-05 | Leonard I Zon |
| National Institutes of Health (NIH) | 5PO1HL32262-32 | Leonard I Zon |
| National Institutes of Health (NIH) | 5R01DK53298 | Leonard I Zon |
| Howard Hughes Medical Institute (HHMI) | | Leonard I Zon |
| Canadian Institutes of Health Research (Insituts de recherche en santé du Canada) | | Michelle I Lin |
| American Heart Association (AHA) | 11POST4920031 | Michelle I Lin |
| Hyundai Hope On Wheels | | Ann Dahlberg |
| National Institutes of Health (NIH) | 1U01HL100395-06 | Irwin D Bernstein |
| Leukemia and Lymphoma Society (LLS) | TRP 6407-13 | Irwin D Bernstein |
| National Institutes of Health (NIH) | K08AR055368 | Richard M White |
| Melanoma Research Alliance (MRA) | Young Investigator Award | Richard M White |
| American Association for Cancer Research (AACR) | Young Investigator Award | Richard M White |
| National Institutes of Health (NIH) | 1R01CA172268 | Cheng Cheng Zhang |

The funders had no role in study design, data collection and interpretation, or the decision to submit the work for publication.

## Author contributions

MIL, Conception and design, Acquisition of data, Analysis and interpretation of data, Drafting or revising the article; ENP, SB, EJH, ET, SS, CWC, AU, AD, Acquisition of data, Analysis and interpretation of data; ADB, SY, Bioinformatics analysis; MCC, Acquisition of data, Contributed unpublished essential data or reagents; ZL, Provided reagents and useful information; CCZ, Provided reagents and useful information, Drafting or revising the article; SHO, IDB, Drafting or revising the article, Contributed unpublished essential data or reagents; JCA, Provided reagents and useful information, Conception and design, Drafting or revising the article; RMW, LIZ, Conception and design, Analysis and interpretation of data, Drafting or revising the article

## Ethics

Animal experimentation: This study was performed in strict accordance with the recommendations in the Guide for the Care and Use of Laboratory Animals of the National Institutes of Health. All zebrafish were housed at the Karp Aquatic Resource Program Facility at Boston Children's Hospital. All protocols were approved by the Animal Care and Use Committee at Boston Children's Hospital and by the Institutional Animal Care and Use Committee (protocol 11-10-2069R).

# Additional files

## Supplementary file

• Supplementary file 1. List of gene sets with significant positive enrichment score from Gene Set Enrichment Analysis (GSEA) of ANGPTL2-stimulated human CD34[+] cells expression data. ANGPTL2 microarray was compared to 5562 gene sets in the Broad MSigDB (c2 curated gene sets, c3 motif gene sets, and c5 GO gene sets). 2521 gene sets showed a positive correlation (i.e., upregulated in ANGPTL2-stimulated cells). 284 gene sets were highly significant with FDR $q$-value of <0.05 and are highlighted in yellow. Those highlighted in red are Myc-related.

## Major datasets

The following datasets were generated:

| Author(s) | Year | Dataset title | Dataset ID and/or URL | Database, license, and accessibility information |
|---|---|---|---|---|
| Lin Michelle I, White Richard M, Zon Leonard I | 2014 | Expression arrays of ANGPTL2-treated CD34 cells | http://www.ncbi.nlm.nih.gov/geo/query/acc.cgi?acc=GSE51652 | Publicly available at the NCBI Gene Expression Omnibus (GSE51652). |
| Lin Michelle I, Trompouki Eirini, Zon Leonard I | 2014 | Notch ChIP-seq from ANGPTL2-treated CD34 cells | http://www.ncbi.nlm.nih.gov/geo/query/acc.cgi?acc=GSE63010 | Publicly available at NCBI Sequence Read Archive (GSE63010). |

Standard used to collect data: The data were collected and reported according to MIBBI Portal for bioscience research.

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
