## [Decision Letter]

Thank you for sending your work entitled “Angiopoietin-like proteins stimulate HSPC development through interaction with Notch receptor signaling” for consideration at *eLife*. Your article has been favorably evaluated by Sean Morrison (Senior editor), a Reviewing editor, and 2 reviewers.

The Reviewing editor and the reviewers discussed their comments before we reached this decision, and the Reviewing editor has assembled the following comments to help you prepare a revised submission.

The reviewers and the Reviewing editor are unanimous in recognizing the quality and importance of the experiments and results presented in this work. But the reviewers have one major objection with the manuscript that needs to be attended to. The reviewers find that the vast array of experiments reported in this manuscript fail to decisively prove the mechanistic details of the model presented. As reviewers are not able to pin-point a single experiment that will resolve this issue, and it is only the authors who know how best to present their data, it is unclear how the reviewers should advise the authors on this manuscript. Suggestions ranged from presenting only the zebrafish data to concentrating on one aspect of the mechanism. It is the policy of *eLife* to either clearly state specific, doable, experiments or not accept the manuscript. Yet, we find that the high quality of the experimental results and the importance of the conclusions, if properly justified, make a compelling case for *eLife* as the appropriate journal for this manuscript. It is the Reviewing editor’s impression that at the core of this indecision is the fact that experiments performed in zebrafish and human cell lines are all intermingled together with conclusions from one applied to results from the other, leaving the reader with the impression that the model is not well substantiated. In some cases, additional justification from unrelated experiments could be trimmed. One possibility is to first state all the results from the zebrafish experiments, filling up any holes with new experiments, if possible and then pointing out, with a schematic diagram, the parts of the pathway that have been uncovered. Then state which steps were either confirmed or gaps in the mechanistic details filled in by experiments in the cell line system. Then create a final compound diagram that distinguishes the contributions from the two systems. Of course, the authors may have a better way to create a more focused manuscript, but it is clear that the reviewers have asked for a more critical analysis of what the data do and do not support; not just a rearrangement of the same paragraphs. Presented below are details of some of the other major concerns, but the authors should attempt to resolve them only if they are willing to also make the mechanistic aspects firmer.

Other specific comments:

1) The conclusions about the stepwise pathways activated are confused by the experiments that show rescue of the *mib ZF* mutants by over expression of either *angptl1/2* (Figure 2) or constitutive AKT (Figure 4). The *mib* mutant impairs the notch pathway by impacting on ubiquitination of endocytosed notch ligands so perhaps this is “upstream” of the dependence of notch cleavage steps on AKT, but the fact that *angptl1/2* over expression rescues the mob phenotype is not further explained or put in context in the Discussion. A final proposed pathway figure would be helpful overall to organize the huge amount of information

2) The results/figures on the embryo injections or rescues by heatshock over expression or crossing show single images and then give ratios (6/6, 7/16 etc.) in the corner of the figure. These are never explained in the legends or Methods, and no statistics are *ever* presented.

3) ChIP-Seq data: It appears that the Notch1 ChIP-Seq did not enrich the RBPJ binding site. This raises questions about the quality of the ChiP-Seq data. As very little is known about the direct targets of Notch1 signaling in human HSCs, what were the positive and negative controls that were used? Additional data about the quality of the ChIP-Seq data needs to be provided.

4) Myc as an important target: Most of the data are correlative. The strongest evidence suggesting that Myc is a direct target of Notch is shown in Figure 6—figure supplement 1 but these studies are performed in human endothelial cells. It is unclear how this relates to the topic of this paper. Were the ChIP-Seq studies revealing for direct Notch regulation of Myc. In addition, the rescue of the *mib* mutant by *myc* expression in Figure 3 is weak, especially compared to the rescue of the angptls-MO cells in Figure 3 or the rescue by ^Myr^hAkt1 in Figure 4–figure supplement 2.

5) Data suggesting that Akt signaling is upstream of Notch: The data certainly support the idea that both Notch and Akt signals are important. However, the studies suggesting that Akt is proximal to Notch are very weak. For example, the effect of Ly294002 on the Notch reporter is much weaker than inhibiting Notch itself (Figure 4). The studies on ADAM7/TACE are very preliminary and in depth functional studies need to be done to evaluate the significance of this result.

6) The data suggesting that the Angptl receptor interacts with Notch are very preliminary: To date, the data on non-canonical Notch ligands has been confusing (at best). For this reason, characterization of non-canonical ligands needs to be rigorous. What is the effect of knocking down the Angptl receptor in zebrafish? Does it affect Notch signaling? Is the interaction specific for Notch1? The data in Figure 7 suggest that the Angptl receptor:Notch interaction is important for Notch signaling but no functional studies are provided to support this. These need to be done. In addition, how does Angptl signaling affect signaling by canonical Notch ligands? Also, the Western blots in Figure 7 are difficult to interpret as they are extremely underexposed.

7) The results in Figure 2 are presented in a complicated manner, and it is not clear if Angptl overexpression in the *Tg(hsp70:zangptl2)* affects *mib* function or *deltaC* expression, or both. It is clear that the hematopoiesis in the embryos is rescued but not clear as to the order of events that lead to the rescue.

---

## [Author Response]

*The reviewers and the Reviewing editor are unanimous in recognizing the quality and importance of the experiments and results presented in this work. But the reviewers have one major objection with the manuscript that needs to be attended to. The reviewers find that the vast array of experiments reported in this manuscript fail to decisively prove the mechanistic details of the model presented*. *[…]*

We thank the editors and reviewers for their comments. We have taken your suggestion in reorganizing the manuscript such that we first present the data obtained from zebrafish followed by data from human cell. We simplified our paper by removing the Akt component as the reviewers pointed out was not as strong and may confuse readers. We have also removed the analysis of *deltaC* expression in the zebrafish embryo (please see response to comment #7). Finally, we have added a more critical analysis of what our data represented and finished with a model figure to demonstrate what we are proposing as the mechanism by which Angptls may regulate Notch.

Other specific comments:

*1) The conclusions about the stepwise pathways activated are confused by the experiments that show rescue of the* mib ZF *mutants by over expression of either* angptl1/2 *(*Figure 2*) or constitutive AKT (*Figure 4*). The* mib *mutant impairs the notch pathway by impacting on ubiquitination of endocytosed notch ligands so perhaps this is “upstream” of the dependence of notch cleavage steps on AKT, but the fact that* angptl1/2 *over expression rescues the mob phenotype is not further explained or put in context in the Discussion. A final proposed pathway figure would be helpful overall to organize the huge amount of information*

We have reorganized our data as stated above to clarify our findings, separating data from zebrafish genetics from human cell. We have also taken out our data on Akt. The work on *akt* illustrated an activation by the *angptls*, but removing it allowed us to tell a simpler story. We will use that data in subsequent publications. Lastly, we have put together a final figure to represent our proposed model so the readers could better understand the activation of *notch* by *angptls*.

*2) The results/figures on the embryo injections or rescues by heatshock over expression or crossing show single images and then give ratios (6/6, 7/16 etc.) in the corner of the figure. These are never explained in the legends or Methods, and no statistics are* ever *presented.*

We appreciate the reviewers for pointing out this omission and have added in our Methods section how the data was quantified.

*3) ChIP-Seq data: It appears that the Notch1 ChIP-Seq did not enrich the RBPJ binding site. This raises questions about the quality of the ChiP-Seq data. As very little is known about the direct targets of Notch1 signaling in human HSCs, what were the positive and negative controls that were used? Additional data about the quality of the ChIP-Seq data needs to be provided*.

Our ChIP-Seq experiments for Notch1 are of high quality. We did find enrichment for the RBPjκ consensus motif but it is embedded as a larger consensus motif for ZNF143. We have clarified this in the text and figure. For Notch ChIP-seq, we have tried several commercially available antibodies but none worked until we tried a published antibody from Jon Aster’s lab. The top motifs from our Notch ChIP-seq, Ets, Runx1 and ZNF143 were similar to those found from his lab (68). Furthermore, we also found enrichment and significant peaks in regions that were also found in his ChIP-seq thus we feel confident of our data.

*4) Myc as an important target: Most of the data are correlative. The strongest evidence suggesting that Myc is a direct target of Notch is shown in*
Figure 6—figure supplement 1
*but these studies are performed in human endothelial cells. It is unclear how this relates to the topic of this paper. Were the ChIP-Seq studies revealing for direct Notch regulation of Myc. In addition, the rescue of the* mib *mutant by* myc *expression in*
Figure 3
*is weak, especially compared to the rescue of the angptls-MO cells in*
Figure 3
*or the rescue by*
^*Myr*^*hAkt1 in Figure 4*–*figure supplement 2.*

Our ChIP-seq for Notch did reveal binding on Myc and this binding region partially overlaps with what has been published in T-ALL (68). We agree that the rescue of *mib* using *myc* mRNA in the now renamed Figure 4 is weaker than those in *angptls-*MO (Figure 4). This was due to some technical difficulties we had with *myc* overexpression, which seemed to be quite toxic to the embryos. The *angptls-*MO morphants are generally healthier than *mib* embryos, probably due to the pantropic deleterious effects of *notch* defects while *angptls-*MO only affected cells that expressed the *zangptl* receptor. Thus we have repeated the experiment many times and this level of rescue is reproducible. We have adjusted the text to reflect the partial rescue and our explanations.

*5) Data suggesting that Akt signaling is upstream of Notch: The data certainly support the idea that both Notch and Akt signals are important. However, the studies suggesting that Akt is proximal to Notch are very weak. For example, the effect of Ly294002 on the Notch reporter is much weaker than inhibiting Notch itself (*Figure 4*). The studies on ADAM17/TACE are very preliminary and in depth functional studies need to be done to evaluate the significance of this result*.

We agree that the studies on Akt are not as conclusive. Given the complexity of the manuscript that was pointed out by the reviewers, we elected to remove the Akt data from our revised manuscript.

*6) The data suggesting that the Angptl receptor interacts with Notch are very preliminary: To date, the data on non-canonical Notch ligands has been confusing (at best)*. *For this reason, characterization of non-canonical ligands needs to be rigorous. What is the effect of knocking down the Angptl receptor in zebrafish? Does it affect Notch signaling? Is the interaction specific for Notch1?*

We are not claiming that Angptls are non-canonical Notch ligands since that would imply Notch activation independent of Notch ligands. Our data simply suggests an additional layer of regulation on Notch afforded by Angptls. We have revised our manuscript to reiterate this point. The zebrafish *angptl* receptor has not been identified. The human LILRB2 belongs to the Ig superfamily, which encompasses more than 850 members. Within this family, LILRB2 is found to be encoded in a large leukocyte receptor complex (LRC) region of chromosome 19, comprising of *KIRs*, *LILRs*, *CEACAMs*, *PSGs*, *Siglec* genes and other extended LRC genes, all of which shared highly similar structure, containing anywhere from 1 to 4 Ig domains in the extracellular region and 1-3 ITAM/ITIM domains intracellularly. Because they are all clustered on 19q13.4, we did extensive bioinformatics searches to find the zebrafish orthologue for LILRB2 but could not discern between the highly similar family members. When we scanned this gene region for synteny, we also could not find the LILRB2 orthologue as a large portion of the region where it could potentially reside within the zebrafish genome seemed to be translocated or deleted. We understand that it would help better interpret our data if we identified the LILRB2 orthologue but we feel that it is beyond the scope of this manuscript. We have highlighted our text in the manuscript about the current lack of a zebrafish receptor.

*The data in*
Figure 7
*suggest that the Angptl receptor:Notch interaction is important for Notch signaling but no functional studies are provided to support this. These need to be done. In addition, how does Angptl signaling affect signaling by canonical Notch ligands? Also, the Western blots in*
Figure 7
*are difficult to interpret as they are extremely underexposed*.

We agree that it would be more conclusive if we can disrupt the interaction between LILRB2 and Notch but because it is not known where the interacting regions between these receptors are, there are currently no blocking antibodies or drugs available that can disrupt this interaction to assess functional loss. We do however show in the now renamed Figure 6 that knocking down LILRB2 prevented the Angptl2-induced Notch cleavage, suggesting that Angptl-mediated Notch activation is at least partially through LILRB2 signaling. We have examined the expression of Notch ligands on the cell membrane in our cultured human endothelial cells and found high levels of Jag1 and lower levels of Jag2, DLL1 and DLL4, which did not seem to change when stimulated by Angptl2. To examine the effect of Angptl2 on signaling by canonical Notch ligands would require disruption between Notch and each of these ligands (by knockdown or specific blocking antibodies). While this is very interesting and adds to our understanding of Angptls signaling, we feel that this is also beyond the scope of the manuscript.

*7) The results in*
Figure 2
*are presented in a complicated manner, and it is not clear if Angptl overexpression in the* Tg(hsp70:zangptl2*) affects* mib *function or* deltaC *expression, or both. It is clear that the hematopoiesis in the embryos is rescued but not clear as to the order of events that lead to the rescue.*

We have revised our manuscript to remove the data on *deltaC* in these rescue experiments in order to simplify our paper. *DeltaC* was examined in our original study because its normal expression overlaps with *notch* in the dorsal aorta during the formation of the hemogenic endothelium (at 28hpf), preceding the formation of HSPCs (36hpf). It also is a *notch* target gene. We hypothesized that since *angptls* may act early, we wanted to see if *deltaC* expression was also restored. Because the gene expression of *deltaC* during development can be regulated by a genetic loop of *notch* pathway self-activation (Holley et al, 2002 [Her1 and the Notch Pathway Function within the Oscillator Mechanism that Regulates Zebrafish Somitogenesis, Development 129:1175-83]), we viewed this rescue of *deltaC* expression as a marker for restored *notch* circuitry in *mib* by *zangtl2* overexpression. But the use of *deltaC* as a marker does raise the question of whether this ligand is involved during *angptl-*mediated *notch* activation. We understand that this may confuse the readers and have now included this only in our Discussion.